# TRANSFORMER-SQUARED: SELF-ADAPTIVE LLMS

**Qi Sun**[1,2*], **Edoardo Cetin**[1*], **Yujin Tang**[1*]
[1]Sakana AI, Japan    [2]Institute of Science Tokyo, Japan
{qisun,edo,yujintang}@sakana.ai
*Equal contribution

## ABSTRACT

Self-adaptive large language models (LLMs) aim to solve the challenges posed by traditional fine-tuning methods, which are often computationally intensive and static in their ability to handle diverse tasks. We introduce Transformer$^2$ (Transformer-Squared), a novel self-adaptation framework that adapts LLMs for unseen tasks in real-time by selectively adjusting only the singular components of their weight matrices. During inference, Transformer$^2$ employs a two-pass mechanism: first, a dispatch system identifies the task properties, and then task-specific "expert" vectors, trained using reinforcement learning, are dynamically mixed to obtain targeted behavior for the incoming prompt. Our method consistently outperforms ubiquitous approaches such as LoRA, with fewer parameters and greater efficiency. Furthermore, Transformer$^2$ demonstrates versatility across different LLM architectures and modalities, including vision-language tasks. Transformer$^2$ represents a significant leap forward, offering a scalable, efficient solution for enhancing the adaptability and task-specific performance of LLMs, paving the way for truly dynamic, self-organizing AI systems. We provide our full source code at https://github.com/SakanaAI/self-adaptive-llms.

## 1    INTRODUCTION

Self-adaptive large language models (LLMs) would represent a significant advancement in artificial intelligence, providing a framework where models can adjust to varied tasks and dynamic contexts in real time. This concept draws inspiration from the longstanding idea of neural networks modifying their own weights to adapt to tasks dynamically (Schmidhuber, 1993; Irie et al., 2022) and neural networks generating weights for other networks, as popularized by HyperNetworks and related methods (Ha et al., 2017; Stanley et al., 2009). While compositionality and scalability are crucial for effective adaptation, current LLM training methodologies fall short of achieving both these properties simultaneously. Our research aims to present a pioneering solution to realize this vision and address these gaps.

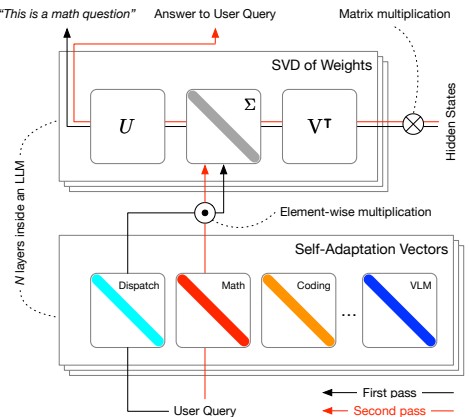

Figure 1: **Overview of Transformer$^2$.** In the training phase, we tune the scales of the singular values of the weight matrices to generate a set of "expert" vectors, each of which specializes in one type of tasks. In the inference phase, a two-pass process is adopted where the first applies the task-specific expert and the second generates the answer.

Traditionally, LLM post-training has sought to optimize a model for a wide range of capabilities in a single, extensive training session. While this "one-shot" fine-tuning framework is ideal from a simplicity perspective, it is also difficult to achieve in practice. For instance, post-training is still resource-intensive. Additionally, expanding data breadth often creates performance trade-offs, making it difficult to simultaneously address overfitting and task interference.

In contrast, self-adaptive models offer a more flexible and efficient approach. Rather than attempting to train an LLM for all tasks in one step, expert modules can be developed offline and augmented to the base LLM on-demand (Kang et al., 2024). This allows the model to dynamically modify its behavior based on the task at hand, without the need for constant re-tuning. In addition to the benefit of having independent components, this modularity also supports continual learning, enabling the model to add new skills over time without catastrophic forgetting. Moreover, self-adaptive LLMs mirror a well-established principle in neuroscience and computational biology, where the brain activates specific regions depending on the task at hand (Loose et al., 2017) and dynamically reconfigures its functional networks in response to changing task demands (Davison et al., 2015).

In principle, the first step toward achieving self-adaptive LLMs can be realized through the development of specialized expert modules, each fine-tuned (Kaplan et al., 2020) via techniques such as low-rank adaptation (LoRA) (Hu et al., 2021). These expert modules can then be dynamically composed at runtime based on the task demands, a process that can be efficiently managed through Mixture of Experts (MoE)-like systems (Tianlong et al., 2024). However, several challenges need to be addressed to make this approach both scalable and compositional. First, fine-tuning LLMs to create multiple expert modules significantly increases the number of parameters that need to be trained. In practice, even with parameter-efficient methods like LoRA, the cumulative size of these modules can quickly escalate, leading to increased storage and computational demands. Second, these expert modules are often prone to overfitting, a phenomenon especially prevalent when training on smaller datasets or narrow task domains. Third, the flexible composition of these expert modules also presents largely unresolved challenges currently posing as open research problems.

To overcome these limitations, we first propose Singular Value Fine-tuning (SVF), a novel parameter-efficient fine-tuning (PEFT) method to obtain effective building blocks for self-adaptation. SVF works by extracting and tuning only the singular values within the model's weight matrices. By focusing on this principled parameterization, our approach mitigates the risk of overfitting, drastically reduces computational demands, and allows for inherent compositionality. We show these properties enable us to cheaply obtain a set of effective domain-specific "expert" vectors by training on narrow datasets with RL, directly optimizing task performance on individual topics.

We then introduce our full Transformer[2] (Transformer-Squared) framework to empower LLMs through the underlying principles of self-adaptation. Given a prompt from an unknown task, Transformer[2] entails a two-pass inference mechanism which we illustrate in Figure 1. During the first pass, Transformer[2] executes the model and observes its test-time behavior, gathering the relevant information to understand the necessary skills to tackle the current problem. During the second pass, our framework uses this information to combine the available expert vectors and provide a new modification to the base weights of the LLM specifically tailored to its test-time conditions. We design three different adaptation strategies that can be used within Transformer[2], which we show provide monotonic performance benefits with increasing access to the test-time conditions.

We evaluate SVF and the full Transformer[2] framework through extensive experiments across a diverse range of LLMs and tasks. First, when trained on domain-specific datasets, we show that SVF consistently outperforms traditional strategies for efficient fine-tuning such as LoRA, and at the same time, with orders of magnitudes fewer parameters. Then we show that Transformer[2] is able to push performance far further, effectively adapting the weights of the base model even in entirely out-of-distribution applications such as visual question answering. Finally, we analyze the properties of our new framework, validating that it provides increasing benefits with additional access to its current test-time conditions and even allow for recycling pre-trained SVF experts across model architectures. In summary, our key technical contributions are the following:

- The development of Transformer[2] as a pivotal self-adaptation framework for LLMs, providing a universal blueprint to dynamically adapt the behavior of LLMs from a growing set of pre-trained skills.
- The introduction of SVF, a novel PEFT method trainable with RL on small datasets, producing compact expert vectors with inherent compositionality, all key properties necessary for our scalable self-adaptation framework.
- The implementation of three adaptation strategies within Transformer[2], effectively dispatching SVF-trained experts with properties designed to cope with different requirements and deployment scenarios.

## 2 RELATED WORKS

**Self-adaptive LLMs** We define self-adaptive LLMs as a group of LLMs or a standalone LLM that can evaluate and modify its behavior in response to changes in its operating environment or internal state, without external intervention. This dynamic adjustment has parallels to concepts like fast-weight memories, which enable networks to update weights in response to task demands (Schmidhuber, 1992; Gomez & Schmidhuber, 2005), and neural network weights being treated as dynamic programs (Schmidhuber, 2015). Recently, Panigrahi et al. (2023) introduces an approach where a smaller auxiliary transformer is updated dynamically within a larger model, aligning with the principles of self-adaptive behavior.

This adaptation can be explored from two perspectives: a macroview, where multiple LLMs collaborate and/or compete, and a microview, where internal adaptations allow a single LLM to specialize in different tasks.

*Macroview:* From this perspective, the system directs queries to LLMs with domain specific expertise, prioritizing outputs from expert models, thereby achieving higher accuracy and task-specific optimization. Such task-specific ensembles can be realized through various mechanisms: multiple LLMs playing distinct roles and coordinate toward a shared goal (Zhuge et al., 2023), engaging in mutual listening and debate (Du et al., 2023), or using meticulously crafted prompt constructions (Zhang et al., 2024) to integrate knowledge library and skill planning. Naturally, the improvement in the specialization and adaptive capabilities of individual LLMs in the ensemble enhances the collective performance. Thus, in this paper, we focus on the microview of self-adaptive LLMs.

*Microview:* MoE in LLMs plays a critical role in this perspective (Tianlong et al., 2024). In MoE systems, inputs are dynamically routed to a subset of specialized modules or layers (e.g., MLPs) containing domain-specific knowledge (Rajbhandari et al., 2022; Fedus et al., 2022). To reduce inference time, researchers introduce sparsely activated MoE where only a subset of the experts are selected per token Jiang et al. (2024); Qwen Team (2024). While it is possible to view Transformer[2] loosely as a type of MoE, there are two major differences. In the aforementioned systems, self-adaptation is achieved through token-level routing, whereas Transformer[2] employs a sample-level module selection strategy. The second difference lies in the construction of expert modules. In traditional MoE systems, expert modules are either trained from scratch (Fedus et al., 2022; Jiang et al., 2024) or dense models (e.g., upcycling) (Qwen Team, 2024; Zhu et al., 2024), without an auxiliary loss to ensure module specialization. In contrast, Transformer[2] specifically trains expert vectors with RL to acquire domain specific-knowledge, making them true experts.

**Low-rank adaptation** PEFT methods such as LoRA (Hu et al., 2021) works by freezing the original model's parameters and introducing small trainable low-rank matrices for task-specific updates. It significantly lowers the computational and memory costs while providing performance comparable to full fine-tuning. Inspired by LoRA's design, various modifications have been proposed (Zhang et al., 2023; Kopiczko et al., 2023; Liu et al., 2024; Bałazy et al., 2024; Cetoli, 2024; Kaushik et al., 2025). Transformer[2] does not rely on low-rank matrices, and instead scales the singular vectors of the original parameter matrix that span the full rank space.

**SVD for LLM Fine-tuning** SVD is increasingly being used as an inductive bias for PEFT in LLMs. For example, Wang et al. (2024) decompose a weight matrix and use the minor singular components, associated with noisy or long-tail information, to initialize low-rank matrices for LoRA fine-tuning. Earlier work proposed using compressed forms like DCT coefficients for generating weight matrices in neural networks (Koutnik et al., 2010), offering efficiency in memory-constrained environments, which resonates with our approach. In a similar vein, SVD is employed to approximate an original weight matrix with the top $r$ singular vectors, corresponding to the highest singular values. A small trainable matrix is then introduced on top of the truncated singular value matrix to adjust the magnitude and orientations within this top-$r$ subspace (Bałazy et al., 2024; Cetoli, 2024). However, the drawback of this approach is that retaining only the top singular components can result in the loss of important information, particularly when the singular values distribution is less skewed. The work most similar to ours is a concurrent effort by Lingam et al. (2024), where they introduce various sparsification methods that utilize the SVD of the weights. However, it is not for self-adaptive LLMs and does not use RL to enhance learning efficiency.

## 3 METHODS

### 3.1 PRELIMINARIES

**Singular value decomposition (SVD)** offers a fundamental view of matrix multiplications. In the context of neural networks, each weight matrix $W \in \mathbb{R}^{n \times m}$ can be decomposed into three components $W = U\Sigma V^\mathsf{T}$, yielding semi-orthogonal matrices $U \in \mathbb{R}^{m \times r}$ and $V \in \mathbb{R}^{n \times r}$ together with an ordered vector of $r$ singular values (in descending order) arranged in the diagonal matrix $\Sigma \in \mathbb{R}^{r \times r}$. The linear operation defined by applying $W$ onto $x$, can be then decomposed into a sum of independent terms, derived from mapping each column $v_i$ from $V$ into the corresponding column $u_i$ from $U$ as $y = \sum_{i=1}^{r} \sigma_i u_i v_i^\mathsf{T} x$. Hence, each singular component represented by the rank-1 matrix $u_i v_i^\mathsf{T}$ independently processes the input, providing an orthogonal contribution to the layer's outputs, with the singular values $\sigma_i$ modulating the degree of the contributions.

**Cross-entropy method (CEM)** is a Monte Carlo method for importance sampling and optimization (Rubinstein & Kroese, 2004). The method is based on the concept of minimizing the KL divergence between two probability distributions $D_{\mathrm{KL}}(P\|Q)$, where $P$ is the target distribution and $Q$ is a maintained distribution. At its core, CEM repeatedly generates a set of samples from $Q$, evaluates these samples with a performance function, and then updates the distribution $Q$ with the characteristics of the elite samples that have performed best. In the standard setup employed in most applications, $Q$ is set to a diagonal multivariate Gaussian, reducing the problem to simply estimating the empirical mean and standard deviation of the latest elites until a stopping criterion is met. We illustrate a complete CEM step in the Python pseudocode below.

```python
def cem_step(mu, sigma, num_elites, num_samples):
    samples = np.random.normal(loc=mu, scale=sigma, size=num_samples)
    scores = evaluate(samples)
    elites = samples[np.argsort(scores)[-num_elites:]]
    new_mu = np.mean(elites, axis=0)
    new_sigma = np.std(elites, axis=0)
    return (new_mu, new_sigma)
```

### 3.2 TRANSFORMER[2]

The construction of Transformer[2] comprises two main steps, for which we provide an illustrative overview in Figure 2. First, we introduce Singular Value Fine-tuning (SVF), a method to learn with RL compact and *compositional* expert vectors based on the SVD of the base model's weights. Then, we describe three different adaptation strategies within Transformer[2], inspired by three orthogonal principles, which adaptively combine the SVF-trained expert vectors during inference. We motivate how the properties of SVF are highly complementary to our adaptation strategies, making Transformer[2] an effective and scalable framework for the design of new self-adaptive LLMs.

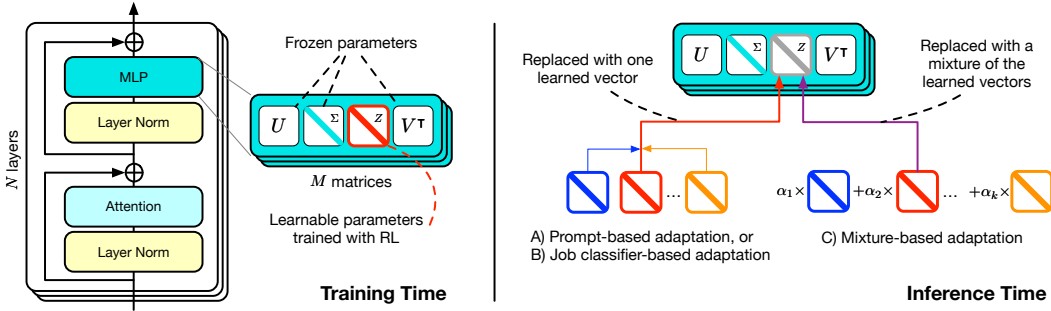

Figure 2: **Method overview.** Left) At training time, we employ SVF and RL to learn the "expert" vectors $z$'s that scale the singular values of the weight matrices. Right) At inference time, we propose three distinct methods to adaptively select/combine the learned expert vectors.

**Singular value fine-tuning** is a key building block in Transformer[2]. It offers an extremely efficient parameterization for fine-tuning and provides inherent compositionality for adaptation. Conven-

tional fine-tuning techniques often aim to augment pre-trained models with new capabilities by modifying their weight matrices. However, in large-scale transformers, these weights are already rich repositories of abstracted knowledge, thanks to the breadth of the pre-training data and expansive architectural design. In fact, as evidenced in much of the prior literature, the requisite capabilities for solving many downstream tasks appear to already exist within these pre-trained models (Sharma et al., 2023). Therefore, instead of seeking to add new features, an efficient fine-tuning approach should focus on making these latent capabilities more expressible. Motivated by these considerations, for any weight matrix $W$, SVF learns a simple vector $z \in \mathbb{R}^r$ that provides targeted modifications to each singular component of $W$ independently, yielding a new weight matrix $W' = U\Sigma'V^\intercal$, where $\Sigma' = \Sigma \otimes \text{diag}(z)$. This essential parameterization enjoys several benefits:

*Negligible parameters:* Learning only a vector $z$ for each weight matrix allows for very efficient fine-tuning with orders of magnitudes fewer optimized parameters even when compared to prior approaches specifically designed for efficiency. For example, the widely popular LoRA approach requires $(m+n) \times r'$ learnable parameters per weight matrix, where $r'$ is a hyper-parameter that generally needs to be set large enough for expressivity. While recent extensions, such as LoRA-XS (Bałazy et al., 2024), try to push efficiency even further, they often introduce limiting assumptions that curb applicability in several practical scenarios (see examples in Appendix C). In contrast, while SVF only needs $r = \min(m, n)$ parameters, we show it empirically does not display the same shortcomings thanks to working on a highly-meaning space provided by the latent expressiveness compressed in the weights of modern LLMs. SVF's scaling only the singular values may seem to lead to limited expressiveness, we wish to point out that the ability to affect the weight matrix in a full-rank manner technically provides more information than low-rank approaches.

*High compositionality:* Decomposing the weights in independent singular components makes the learned $z$ vectors highly composable and interpretable, opening numerous possibilities for adaptation via algebraic manipulations. Instead, LoRA-based methods inherently lack these properties. For instance, even if two LoRAs learned on the same task were to learn exactly the same adjustments for each $W$, directly interpolating between their compressed $A$ and $B$ matrices is unlikely to preserve any of their original behavior, given the countless number of equivalent parameter permutations they might have converged to.

*Principled regularization:* Exclusively modifying the magnitude of pre-existing singular components provides a principled and effective form of regularization. In practice, this property enables us to fine-tune for arbitrary downstream tasks with only hundreds of data points without the risk of severe collapse or overfitting.

**End-to-end optimization with RL.** We train a set of SVF vectors $\theta_z = \{z_1, \cdots, z_{N \times M}\}$ to fine-tune an arbitrary language model $\pi_{\theta_W}$ parameterized by $\theta_W$ with RL, optimizing directly for task performance. Here, $\theta_W = \{W_1, \cdots, W_{N \times M}\}$ is the set of weight matrices, where $N$ is the number of layers and $M$ is the number of weight matrices to fine-tune per layer. We use the seminal REINFORCE algorithm (Williams, 1992) and label each generated answer $y_i$ (for the prompt $x_i \in D$) with a unitary reward based on its correctness $r \in \{-1, 1\}$. Inspired by related applications of RL for optimizing LLMs (Ouyang et al., 2022), we regularize the REINFORCE objective by adding a KL penalty for deviating from the original model's behavior, weighted by a small coefficient $\lambda \in \mathbb{R}^+$. Thus, our final objective function can be written as:

$$J(\theta_z) = \mathbb{E}\left[\log\left(\pi_{\theta_{W'}}(\hat{y}_i \mid x_i)\right) r(\hat{y}_i, y_i)\right] - \lambda D_{\text{KL}}(\pi_{\theta_{W'}} \| \pi_{\theta_W}), \tag{1}$$

where we use $\pi_{\theta_{W'}}$ to denote the resulting language model after substituting the original weight matrices $W$ with $W'$. While RL is generally considered less stable than next-token prediction objectives, we find the regularization properties of SVF avoid many of the failure modes of prior less-constrained parameterizations (see Section 4.3). Thus, combining these complementary components effectively enables us to avoid relying on expensive fine-tuning procedures with large hand-designed datasets as proxies, and directly maximize task performance end-to-end.

In general, SVF with RL puts lower requirement on the dataset it trains on. For example, LoRA fine-tuning requires "explaining texts" to perform next token predictions, which puts a higher requirement on the dataset (e.g., imagine LoRA fine-tuning on a GSM8K dataset where no reasoning text but only the final number is provided). This benefit allows SVF to be more general and effective. One possible caveat SVF can face is the sparse rewards caused by a weak base model, which we discuss this further in Section 5.

**Self-adaptation** is a critical mechanism in nature that has established itself as a core guiding principle in modern system design (Klös et al., 2015). Our initial efforts toward self-adaptive foundation models focus on the inference stage of LLMs, where we devise a simple two-pass adaptation strategy that combines $K$ sets of base "expert" vectors $z^{1:K}$ trained with SVF to provide different kinds of capabilities (e.g., coding, math, etc). The mapping between a capability and the dataset we train on can be acquired in the dataset's meta data. In the first inference pass, given a task or an individual input prompt, Transformer[2] executes the model and observes its test-time behavior to derive a new $z'$ vector tailored to its test-time conditions. This adapted $z'$ is then used in the second inference pass to provide an actual response with the newly adapted weights. The interaction between SVF-trained expert vectors and the adaptation strategies ensures seamless integration, where expert vectors provide modular capabilities, and the adaptation strategies dynamically determine and compose the most suitable combination to address the input task. In this first work, we propose three simple approaches to produce the vector $z'$ during the first inference pass, implementing self-adaption with distinct methods and requirements. Below, we provide an outline of each method and refer to Appendix A for additional implementation details.

*A) Prompt engineering:* Our most basic approach involves constructing a new "adaptation" prompt which we use to directly *ask* the LLM to categorize the input prompt. Based on its response, we then extract one category out of the set of domain topics used to pre-train each SVF expert and, thus, we select the corresponding $z'$ directly from $z^{1:K}$. In our adaptation prompt, we also explicitly provide the option for a generic "others" category, allowing the model to use its base weights in case no expert provides appropriate capabilities. We show the format used to construct the adaptation prompt in Figure 3.

*B) Classification expert:* A direct extension of the prompt engineering approach comes from using a specialized system to handle task identification. Following the principles of self-adaptation, we apply SVF to fine-tune the base LLM itself to handle this task. In particular, we collect a dataset $D = \{(x_{1,1}, 1), \cdots, (x_{i,k}, k), \cdots\}$ from the $K$ SVF training tasks, where $x_{i,k}$ is the $i$-th example from the $k$-th expert task. Each tuple $(x_{i,k}, k)$ then forms an example to pre-train an additional job classification expert $z^c$ learned in the same fashion as the others. During the first inference pass, we simply load $z^c$, intending to improve the inherent task classification capabilities of the base model to select a more appropriate $z'$ to handle the input prompt.

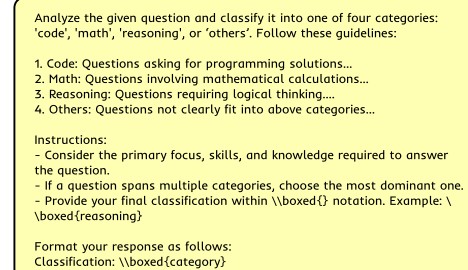

Analyze the given question and classify it into one of four categories: 'code', 'math', 'reasoning', or 'others'. Follow these guidelines:

1. Code: Questions asking for programming solutions...
2. Math: Questions involving mathematical calculations...
3. Reasoning: Questions requiring logical thinking....
4. Others: Questions not clearly fit into above categories...

Instructions:
– Consider the primary focus, skills, and knowledge required to answer the question.
– If a question spans multiple categories, choose the most dominant one.
– Provide your final classification within \\boxed{} notation. Example: \\boxed{reasoning}

Format your response as follows:
Classification: \\boxed{category}

Figure 3: **Prompt based adaptation.** Self-adaptation prompt used by Transformer[2] to classify the task prompt into pre-defined categories.

*C) Few-shot adaptation:* Our third approach leverages additional task information by assuming extended access to its test-time conditions beyond individual prompts. Our approach is inspired by popular few-shot prompting techniques, which have been shown to provide consistent performance improvements and even allow LLMs to "in-context" learn tasks that were entirely unseen prior to inference (Brown, 2020). For each optimized $W$, our approach entails producing an entirely new $z' = \sum_{k=1}^{K} \alpha_k z_k$ by linearly interpolating between the $K$ learned SVF vectors, each weighted by the coefficients $\alpha_k$. We employ CEM to search over the possible values of each $\alpha_k$ based on the performance on a set of "few-shot prompts", which are specifically held out from the rest of the test prompts and used to evaluate CEM's population samples. In the case of multiple population samples obtaining the same score on these held-out prompts, we break ties by favoring the one with the highest average log-likelihood across its own generated correct answers. Crucially, we only need to perform this process once for each target task, avoiding the need to increase the length of each question prompt, a relevant downside of traditional few-shot prompting. We refer to Section A.4, for additional details and an extended discussion of this final approach.

## 4 EXPERIMENTS

We extensively evaluate Transformer[2] on multiple tasks and models with the purpose of: (1) assessing the efficiency and effectiveness of SVF; (2) demonstrating self-adaptiveness through the

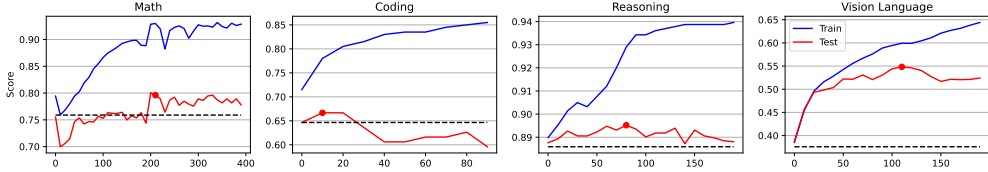

Figure 4: **SVF learning curves.** The dashed lines indicate the performance of LLAMA3-8B-INSTRUCT on the test split of each task. SVF effectively fine-tunes to surpass the base performance. While we use the best validation score to select our checkpoint for evaluation (marked by red dots), we present the entire training curve without early stopping to demonstrate SVF's learning capabilities. Tasks with only hundreds of training samples like Coding and Reasoning were stopped early. In our experiments, we update the parameters at the end of each epoch.

three proposed adaptation strategies; (3) conducting in-depth analysis and ablation studies aimed at understanding and interpreting the properties of our new framework.

## 4.1 EXPERIMENTAL SETUPS

To validate the generality of Transformer[2] we consider three pre-trained LLMs ranging across different model families and architecture sizes: LLAMA3-8B-INSTRUCT, MISTRAL-7B-INSTRUCT-V0.3, and LLAMA3-70B-INSTRUCT. For each model, we obtain three sets of SVF-trained $z$ vectors to maximize performance for GSM8K (Cobbe et al., 2021), MBPP-pro (Austin et al., 2021), and ARC-Easy (Clark et al., 2018), respectively. Additionally, we also train a set of $z$ vectors for LLAMA3-8B-INSTRUCT, when applied as the language backbone for TextVQA (Singh et al., 2019), in order to assess SVF's applicability to the vision-language modeling (VLM) domain. We provide SVF's main learning curves on each of these tasks in Figure 4. Finally, we evaluate the full Transformer[2] adaptation framework on four unseen tasks: MATH (Hendrycks et al., 2021), Humaneval (Chen et al., 2021), ARC-Challenge (Clark et al., 2018), and OKVQA (Marino et al., 2019). In all our adaptation experiments, we only consider experts obtained in the pure-language settings, assessing its test-time applicability even for the distinctive vision domain. Please refer to the Appendix A for additional details and a summary of the hyper-parameters used in the experiments.

## 4.2 EXPERIMENTAL RESULTS

**SVF performance** We provide results after training on each considered task with the LLAMA3-8B-INSTRUCT, MISTRAL-7B-INSTRUCT-V0.3, and LLAMA3-70B-INSTRUCT base models in Table 1. Remarkably, we find that SVF provides considerable and consistent performance gains across nearly all tasks and base models. Instead, LoRA experts yield smaller gains and even sporadic performance degradation. (These LoRA experts are trained with next token prediction. While we also have LoRA experts trained with RL in Table 4, RL seems work less well with LoRA than with SVF.) This observed trend extends also to the vision-language domain, as fine-tuning LLAMA3-LLAVA-NEXT-8B with SVF bolsters the base model's performance by over 39% (see Figure 5). To ensure a fair comparison, we provide extensive ablations to both our model and the LoRA baseline considering different architecture and optimization objectives in Appendix 4.3). Due to its essential

Table 1: **Fine-tuning results.** LLM performance on the test splits of math, coding and reasoning. Normalized scores are in the parentheses.

| Method | GSM8K | MBPP-Pro | ARC-Easy |
|---|---|---|---|
| LLAMA3-8B-INSTRUCT | 75.89 (1.00) | 64.65 (1.00) | 88.59 (1.00) |
| + LoRA | 77.18 (1.02) | **67.68 (1.05)** | 88.97 (1.00) |
| + SVF (Ours) | **79.15 (1.04)** | 66.67 (1.03) | **89.56 (1.01)** |
| MISTRAL-7B-INSTRUCT-V0.3 | 42.83 (1.00) | 49.50 (1.00) | 81.65 (1.00) |
| + LoRA | 44.66 (1.04) | 51.52 (1.04) | 81.19 (0.98) |
| + SVF (Ours) | **49.74 (1.16)** | **51.52 (1.04)** | **85.14 (1.04)** |
| LLAMA3-70B-INSTRUCT | 85.29 (1.00) | **80.81 (1.00)** | **89.10 (1.00)** |
| + LoRA | 77.26 (0.91) | 68.69 (0.85) | 88.55 (0.99) |
| + SVF (Ours) | **88.32 (1.04)** | **80.81 (1.00)** | 88.47 (0.99) |

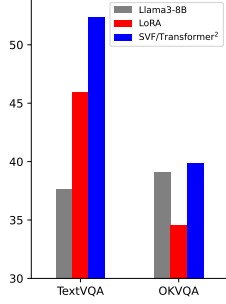

Figure 5: **Results for the VLM domain.**

Table 2: **Self-adaptation on unseen tasks.** Normalized scores are in the parentheses.

| Method | MATH | Humaneval | ARC-Challenge |
|---|---|---|---|
| LLAMA3-8B-INSTRUCT 3 | 24.54 (1.00) | 60.98 (1.00) | 80.63 (1.00) |
| + LoRA | 24.12 (0.98) | 52.44 (0.86) | 81.06 (1.01) |
| + Transformer$^2$ (Prompt) | 25.22 (1.03) | 61.59 (1.01) | 81.74 (1.01) |
| + Transformer$^2$ (Cls-expert) | 25.18 (1.03) | 62.80 (1.03) | 81.37 (1.01) |
| + Transformer$^2$ (Few-shot) | **25.47 (1.04)** | **62.99 (1.03)** | **82.61 (1.02)** |
| MISTRAL-7B-INSTRUCT-v0.3 | 13.02 (1.00) | 43.29 (1.00) | 71.76 (1.00) |
| + LoRA | 13.16 (1.01) | 37.80 (0.87) | **75.77 (1.06)** |
| + Transformer$^2$ (Prompt) | 11.86 (0.91) | 43.90 (1.01) | 72.35 (1.01) |
| + Transformer$^2$ (Cls-expert) | 11.60 (0.89) | 43.90 (1.01) | 74.83 (1.04) |
| + Transformer$^2$ (Few-shot) | **13.39 (1.03)** | **47.40 (1.09)** | 75.47 (1.05) |
| LLAMA3-70B-INSTRUCT | **40.64 (1.00)** | 78.66 (1.00) | 87.63 (1.00) |
| + LoRA | 25.40 (0.62) | 73.78 (0.94) | 83.70 (0.96) |
| + Transformer$^2$ (Prompt) | 40.44 (1.00) | **79.88 (1.02)** | **88.48 (1.01)** |

parameterization, we would like to note that training SVF requires considerably fewer resources, with less than 10% of the training parameters of our LoRA implementation.

**Adaptation performance** With the SVF trained $z$ vectors, we assess the self-adaptation capability of Transformer$^2$ on unseen tasks. For a fair comparison with LoRA, we record the performance of this baseline using all checkpoints from the considered training tasks and report only its highest performance for each of the test tasks. As shown in Table 2, all of our Transformer$^2$ adaptation strategies demonstrate improvements across all tasks for LLAMA3-8B-INSTRUCT base models, and in at least two out of three tasks for both MISTRAL-7B-INSTRUCT-v0.3 and LLAMA3-70B-INSTRUCT. In contrast, even the best training LoRAs only provide marginal improvements on the ARC-Challenge task and still significantly deteriorate performance on both MATH and Humaneval. This discrepancy suggests that LoRA's parameterization and optimization might be particularly sensitive to overfitting. In Figure 5, we find a similar dichotomy in the OKVQA task, with the performance of the base LLAMA3-LLAVA-NEXT-8B VLM only improving after applying Transformer$^2$.

Comparing the three proposed adaptation strategies, we highlight a clear monotonic trend – with more involved strategies and information about the test-time condition, self-adaptation appears to be increasingly effective. In particular, Transformer$^2$ with few-shot self-adaptation is almost always the highest-scoring method, providing notable improvements across all settings except for LLAMA3-70B-INSTRUCT @MATH, where we have only SVF-tuned half of the layers due to limited GPU resources. This trend shows that providing additional or different kinds of information seems highly benefit our framework, suggesting that Transformer$^2$ could provide foundation models with new means to continually improve performance when deployed in lifelong settings.

Table 3 reports the inference time required by the prompt adaptation strategy of Transformer$^2$, with the time spent on solving the entire problem set presented separately for the 1st and 2nd passes. Notice that the 2nd pass inference time is the time spent on solving the problems, and the 1st pass inference time is the time for self-adaptation, 1st to 2nd pass inference time ratios are in parentheses. While the additional inference pass appears to double the overall runtime, it is important to note that inference time primarily depends on the number of tokens generated. In our settings, it is $\mathcal{O}(n)$ where $n$ is the length of the input. ARC-challenge have larger cost ratio because they

Table 3: **Time cost of 2-pass inference in prompt adaptation strategy of Transformer$^2$ for the entire problem set.** 1st to 2nd pass inference time ratios are shown in parentheses.

| Task | 1st (s) | 2nd (s) |
|---|---|---|
| MATH | 42.64 (13%) | 321.19 |
| Humaneval | 2.76 (19%) | 14.28 |
| ARC-Challenge | 13.40 (47%) | 28.51 |

are single choice problems and the cost of the 2nd pass is also $\mathcal{O}(n)$. In general settings, we think it is reasonable to assume this ratio to be closer to those of MATH and Humaneval. For a detailed discussion on improving the efficiency of CEM adaptation methods, please see Appendix D

### 4.3 ANALYSIS

Lastly, we analyze and discuss the properties of our adaptation strategies for which we provide extensions and further discussion Appendix B.

Figure 6: **Confusion matrices.** These matrices display the classification percentages, where rows represent the task classes (ground truth) and columns indicate the predicted categories. Some samples are misclassified as "Others," which is reflected in rows where the totals do not sum to one.

**Analysis 1: Job dispatching accuracy** In Figure 6 we provide the confusion matrices of classification-based adaptation strategies. Results validate the effectiveness of both our strategies to match each prompt with experts trained in similar domains, as evidenced by the high values along the diagonals. Furthermore, the results from LLAMA3-8B-INSTRUCT and MISTRAL-7B-INSTRUCT-V0.3 also show that using the classification expert consistently provides higher classification accuracy than vanilla prompt engineering. While this difference could explain the higher performance of the relative self-adaptation strategy, we also note that domain similarity might not be the only metric relevant to identifying the best expert for each prompt or task. To this end, we believe many further extensions could be explored in future work, using heuristics such as past expert performance or token-level analysis to further push our framework's scalability.

**Analysis 2: Training tasks adaptation contribution** In Figure 7, we show the normalized adaptive coefficients $a_k$ interpolating between SVF vectors learned via CEM for LLAMA3-8B-INSTRUCT and MISTRAL-7B-INSTRUCT-V0.3 across the unseen tasks. Intuitively, we find that the expert vectors from the training tasks sharing similar topics to the unseen ones are often the highest contributors. However, we observe that the MATH task appears as an exception, as the $a_k$ for the GSM8K expert is actually the lowest in both models. We hypothesize this reflects the different nature of the mathematics competition problems from MATH as compared to the grade-school problems in GSM8K. In fact, not only is the difficulty of the MATH questions far beyond GSM8K, but a large portion of its problems also hinges mainly on logical reasoning, for which a task like ARC might be more aligned. Furthermore, we note that the different $z$ vectors appear to contribute more uniformly to adaptation in the Llama model. This difference indicates that, due to its higher base performance, the Llama model does not need to rely on any particular set of skills as much as Mistral, and can harness more holistic benefits from self-adaptation. Note that applying $a_k$ uniformly is not a universal solution for leveraging expert vectors. This becomes evident when we look at different model and task combinations (e.g. applying $a_k$ uniformly on LLAMA3-8B-INSTRUCT for MATH tasks only achieves 24.47, while Transformer$^2$ (Few-shot) achieves 25.47).

**Analysis 3: Ablation studies**

*Module sensitivity:* We compare the performance of SVF on different modules (see trials 1-3). Both individual MLP and attention updates improve performance, with MLP resulting in more pronounced gains. Updates to both module types yield even more significant enhancements.

*Objective function:* We are interested in the performance impact from different objective functions, and we compare the RL objective with next-token prediction (see trials 2, 4). For the latter, we use instruction fine-tuning with official GSM8K solutions as target tokens. Results show clear performance gains with RL. Conversely, next-token prediction even hinders performance. This highlights RL's ability to handle cases lacking detailed solutions, suggesting its superiority in this context.

*SVF vs LoRA:* We also evaluate LoRA using the RL objective (see trials 2, 5). A significant performance disparity is observed, primarily attributed to the severe instability of the LoRA training process. Despite exploring a wide range of learning rates, LoRA's performance consistently lagged behind. For further illustrations, see Figure 9 in the appendix.

**Analysis 4: Cross-model compatibility** Finally, we explore the potential for our framework to be applied *across different LLMs*. In particular, we evaluate whether the SVF vectors trained on LLAMA3-8B-INSTRUCT can benefit MISTRAL-7B-INSTRUCT-V0.3, and whether we can perform adaptation across the vectors of these two models. We present our main findings in Table 5 and refer to Appendix B for additional results. Surprisingly, we find that positive transfer occurs with

Table 4: **Ablation studies.** We fine-tune LLAMA3-8B-INSTRUCT on the GSM8K training split with different settings and the results on the test split along with zero-shot transfer results on MATH.

| # | Method | Objective Function | Module | #Params ($\downarrow$) | GSM8K ($\uparrow$) | MATH ($\uparrow$) |
|---|--------|--------------------|--------|------------------------|--------------------|-------------------|
| 0 | | LLAMA-3-8B-INSTRUCT | | | 75.89 (1.00) | 24.54 (1.00) |
| 1 | SVF | Policy gradient | MLP | 0.39M | 78.62 (1.04) | 24.20 (0.99) |
| 2 | SVF | Policy gradient | attention | **0.16M** | 76.19 (1.00) | 24.20 (0.99) |
| 3 | SVF | Policy gradient | MLP + attention | 0.58M | **79.23 (1.04)** | **25.04 (1.04)** |
| 4 | SVF | Next token pred | attention | **0.16M** | 60.50 (0.80) | 18.52 (0.75) |
| 5 | LoRA | Policy gradient | attention | 6.82M | 57.92 (0.76) | 15.72 (0.64) |
| 6 | LoRA | Next token pred | attention | 6.82M | 77.18 (0.98) | 24.12 (0.96) |
| 7 | LoRA | Next token pred | MLP + attention | 35.13M | 75.66 (0.96) | 22.12 (0.91) |

Table 5: **Cross-model $z$ vector transfer.** Results from transferring the expert vectors trained on LLAMA3-8B-INSTRUCT to MISTRAL-7B-INSTRUCT-V0.3 with cross model few-shot adaptation.

| Method | MATH | Humaneval | ARC-Challenge |
|--------|------|-----------|---------------|
| *SVF training task* | *GSM8K* | *MBPP-pro* | *ARC-Easy* |
| MISTRAL-7B-INSTRUCT-V0.3 | **13.02 (1.00)** | 43.29 (1.00) | 71.76 (1.00) |
| + Llama SVF (ordered $\sigma_i$) | 11.96 (0.92) | 45.12 (1.04) | 72.01 (1.00) |
| + Llama SVF (shuffled $\sigma_i$) | 10.52 (0.81) | 40.24 (0.93) | 70.82 (0.99) |
| + Few-shot adaptation (cross-model) | 12.65 (0.97) | **46.75 (1.08)** | **75.64 (1.05)** |

visible benefits in 2 out of 3 tasks. We note these improvements are due to the inherent ordering of the SVF parameterization, as *randomly shuffling* each SVF vector before applying it to the Mistral model consistently degrades performance.

This operation leads to notable performance degradation across tasks. By performing few-shot adaptation using the SVF vectors from both models, the performance of MISTRAL-7B-INSTRUCT-V0.3 further improves across the board. We observe that these gains even surpass the best score from adapting MISTRAL-7B-INSTRUCT-V0.3 with *all* the SVF vectors in the ARC-Challenge task reported in Table 2. While these results appear promising, we note that the surprising compatibility discovered through our naive transfer approach is potentially tied to the similarity between the architectures of the two considered LLMs. To this end, whether similar transfer can be replicated with models of different scales remains an open question that could open the doors to recycling task-specific skills for larger models, with important implications for democratization and sustainability.

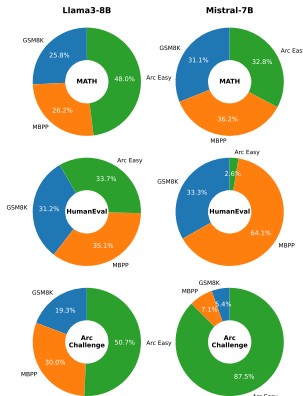

Figure 7: $\alpha_k$ **learned weights.**

## 5 CONCLUSION

In this paper, we introduced Transformer[2], providing a novel blueprint toward realizing self-adaptive LLMs. Within this framework, we first proposed SVF, offering superior performance than prior fine-tuning recipes, together with reduced costs, high compositionality, and overfitting regularization – all crucial properties to achieve scalable adaptation. Leveraging a set of SVF experts as building blocks, we developed three effective strategies for self-adaptation, each offering unique advantages, with performance and monotonic performance benefits with increasing access to the test-time conditions.

While Transformer[2] demonstrates promising results, there remain exciting opportunities for future work. One limitation is that the capabilities of SVF experts are tied to the latent components of the base model. To address this, model merging offers a promising direction (Yu et al., 2024; Goddard et al., 2024; Akiba et al., 2024). Additionally, while CEM-based adaptation balances performance and efficiency, scaling to a large number of specialized domains may increase the initial one-time computational costs. Advances in model merging and efficient adaptation techniques have produced models dominating open leaderboards, making them strong candidates as base models for Transformer[2] and opening new possibilities for adaptive LLMs.

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

## A  IMPLEMENTATION DETAILS AND HYPER-PARAMETERS

### A.1  SVF TRAINING

We obtain the expert vectors $z$ as the base components in Transformer[2] by training the SVF fine-tunes with a consistent recipe across the considered training tasks and language models. We divide each dataset to produce equal-sized training and validation splits. We then apply our RL-based approach, optimizing $\theta_z$ with AdamW using a learning rate of $2 \times 10^{-3}$ with cosine decay, a batch size of 256, and gradient clipping. We employ early stopping and select the best $\lambda$ (the coefficient of the KL divergence term) based on validation performance. For the LLAMA3-70B-INSTRUCT and Vision tasks experiments, we apply the SVF on half of the layers to reduce memory usage while maintaining considerable performance improvement. During the training of LLAMA3-8B-INSTRUCT on the vision language tasks, we apply a small negative reward (-0.1) for training stability.

### A.2  LORA TRAINING

We follow community best practices for LoRA fine-tuning, applying it to query and value projection layers with learning rates around $5 \times 10^{-5}$. We set 200 total iterations with a 256 global batch size for sufficient training. For feasible LoRA instruction training, we collect solutions for all training tasks (GSM8K, MBPP, Arc-Easy, TextVQA) from official sources and append them to question prompts. Table 8 shows a sample math problem used for LoRA fine-tuning. Despite extensive hyperparameter tuning, we often observe test performance decay as discussed, which can be attributed to the small number of training samples and potential model requirements for instruction fine-tuning data (specifically, the highly detailed thinking process).

> Below is an instruction that describes a task. Write a response that appropriately completes the request.
>
> Natalia sold clips to 48 of her friends in April, and then she sold half as many clips in May. How many clips did Natalia sell altogether in April and May?
>
> Natalia sold 48/2 = <<48/2=24>>24 clips in May. Natalia sold 48+24 = <<48+24=72>>72 clips altogether in April and May. #### 72

Figure 8: **Sample problem and answer.** Math data sample used for LoRA instruction fine-tuning, text in blue is the unmasked solution.

### A.3  HYPER PARAMETERS

We present a summary of the hyperparameters used in our experiments in Table 6. To optimize performance, we conducted sweeps across several hyperparameters and selected the most effective combination based on validation results. For SVF, our primary focus was on adjusting the KL coefficient to enhance training stability. In the case of LoRA, we concentrated on sweeping the learning rate and maximum gradient clip norm to identify optimal settings.

### A.4  FEW-SHOT ADAPTATION

As described in the main text, our few-shot adaptation approach entails producing an entirely new $z' = \sum_{k=1}^{K} \alpha_k z_k$ for each $W$ by linearly interpolating between the $K$ learned SVF vectors, each weighted by the coefficients $\alpha \in \mathbb{R}^K$. We employ CEM to search for $\alpha_k$'s based on the performance on the few-shot prompts, which are specifically held out from the rest of the test prompts and used to obtain the elite set at each iteration. In the case of multiple sample solutions obtaining the same score on these held-out samples, we break ties by choosing the sample solution with the highest average log-likelihood across the tokens of its generated correct answers.

In all of our main experiments, we reserve only 10 samples of data for self-adaptation and perform up to 100 CEM iterations. For each setting, we consider both per-layer and per-vector adaptation, where the latter strategy has the advantage of greatly simplifying search (as we only have 3 $\alpha$ coefficients). Moreover, we experiment with both normalizing across the $\alpha$ of different tasks (such that their sum would be fixed to 1) or keeping them unconstrained. Due to the lack of a validation set, we simply report the performance attained by our best sample from these test configurations at the end of optimization, on the remaining unseen samples for each task.

Table 6: **Hyper-parameters used for SVF and LoRA training.** We perform a sweep on certain sensitive hyper-parameters across methods for fair comparison.

| SVF Hyperparameters | |
|---|---|
| Initial mean value of $z$ | 0.1 |
| Initial variance value of $z$ | $1 \times 10^{-3}$ |
| Global batch size | 256 |
| Learning rate | $2 \times 10^{-3}$ |
| Clip max norm | $1 \times 10^{-3}$ |
| KL coefficient $\lambda$ | 0.0, 0.1, 0.2, 0.3 |
| **LoRA Hyperparameters** | |
| Rank | 16 |
| LoRA alpha | 32 |
| LoRA dropout | 0.05 |
| Global batch size | 256 |
| Learning rate | $2 \times 10^{-4}, 5 \times 10^{-4}, 2 \times 10^{-5}, 5 \times 10^{-5}, 2 \times 10^{-6}. 5 \times 10^{-6}$, |
| Clip max norm | $1 \times 10^{-3}, 1.0$ |

Table 7: **Additional Comparison Experiment.** Normalized scores are in the parentheses.

| Method | GSM8K | MBPP-Pro | ARC-Easy |
|---|---|---|---|
| LLAMA3-8B-INSTRUCT | 75.89 (1.00) | 64.65 (1.00) | 88.59 (1.00) |
| + IA3 | 78.01 (1.03) | **67.68 (1.05)** | 89.10 (1.01) |
| + DORA | 78.09 (1.03) | 64.65 (1.00) | 89.14 (1.01) |
| + SVF(Ours) | **79.15 (1.04)** | 66.67 (1.03) | **89.56 (1.01)** |

| Method | MATH | Humaneval | ARC-Challenge |
|---|---|---|---|
| LLAMA3-8B-INSTRUCT | 24.54 (1.00) | 60.98 (1.00) | 80.63 (1.00) |
| + IA3 | 23.64 (0.96) | 59.76 (0.98) | 81.57 (1.01) |
| + DORA | 24.44 (0.99) | 52.44 (0.86) | 81.14 (1.01) |
| + Transformer$^2$ (Prompt) | 25.22 (1.03) | 61.59 (1.01) | 81.74 (1.01) |
| + Transformer$^2$ (Cls-expert) | 25.18 (1.03) | 62.80 (1.03) | 81.37 (1.01) |
| + Transformer$^2$ (Few-shot) | **25.47 (1.04)** | **62.99 (1.03)** | **82.61 (1.02)** |

# B ADDITIONAL RESULTS

## B.1 BASELINE COMPARISON TO MORE PEFT METHODS

We conduct additional comparison studies against more parameter-efficient fine-tuning methods, including IA3Liu et al. (2022), DORA. Liu et al. (2024).

As Table 7 shows, SVF still outperforms other methods and shows promising generalized performance.

## B.2 IMPACT FROM NUMBER OF FEW-SHOTS

We investigate the relationship between the number of samples available for few-shot adaptation and downstream performance. Our analysis focused on the test task where LLAMA3-8B-INSTRUCT demonstrates the highest baseline performance, to prevent the potential for a null signal in our CEM-based search.

Table 8: **Few-shot adaptation scaling on the Arc-Challenge task.** Performance varies with number of examples.

| Method | Transformer$^2$ | IA$^3$ 100 steps | IA$^3$ 1000 steps |
|---|---|---|---|
| LLAMA3-8B-INSTRUCT | 80.63 (1.00) | 80.63 (1.00) | 80.63 (1.00) |
| + 3-shot adaptation | 82.18 (1.02) | 81.83 (1.01) | 79.01 (0.98) |
| + 5-shot adaptation | 82.38 (1.02) | 80.89 (1.00) | 79.41 (0.98) |
| + 10-shot adaptation | **82.61 (1.02)** | **82.00 (1.02)** | **79.78 (0.99)** |
| + 20-shot adaptation | **82.61 (1.02)** | 81.40 (1.01) | 79.61 (0.99) |

As Table 8 shows, substantial benefits of our few-shot strategy are evident with as few as 3 to 5 test samples. Moreover, performance appears to plateau beyond 10 samples, underscoring how our essential and inherently regularized SVF pa-

rameterization effectively complements self-adaptation. This efficiency enables optimal use of data to enhance understanding of the test task.

For completeness, we have also conducted experiments with identical settings on IA[3] (Liu et al., 2022), another method that leverages few-shot examples. All experiments were conducted with full batch size, a learning rate of $5 \times 10^{-5}$, with 100 and 1000 training steps.

Our results indicate that the performance of IA[3] on the unseen test tasks is inferior to CEM-based adaptation for all numbers of few shots considered. We note that in our experiment, we have to considerably limit the number of optimization steps to avoid overfitting the 500,000 parameters of IA[3] on the few-shot samples. However, we believe overfitting might still be occurring to some degree even after only 100 steps, as also validated by the model's perfect training accuracy on this extremely small dataset. This limitation of fine-tuning-based adaptation highlights the superior generalization capability of our CEM-based adaptation approach in Transformer[2].

### B.3 CROSS-MODEL SVF TRANSFER ON THE TRAINING TASKS

We provide complementary results to Table 5 in the main text, where we analyze the SVF cross-model transfer performance from training on GSM8K, MBPP-pro, and ARC-Easy to our considered test tasks. In Table 9, we show the results in the same transfer setting this time evaluating MISTRAL-7B-INSTRUCT-V0.3 on the same training tasks where the LLAMA3-8B-INSTRUCT SVF vectors were obtained from. Overall, we recognize a similar trend, albeit with less consistent improvement from the original model (only in 1 out of 3 tasks), but still much higher performance than the randomly shuffled baseline. These results further confirm that the canonical ordering of the SVF parameterization is key for cross-model transfer, highlighting once more its inherent suitability to empower self-adaptation.

Table 9: **Cross-model $z$ Vector Transfer.** Results from transfering the SVF expert vectors trained on LLAMA3-8B-INSTRUCT to MISTRAL-7B-INSTRUCT-V0.3 in the respective training tasks.

| Method | GSM8K | MBPP-pro | ARC-Easy |
|---|---|---|---|
| MISTRAL-7B-INSTRUCT-V0.3 | **42.83** (1.00) | **49.50** (1.00) | 81.65 (1.00) |
| + Llama SVF (ordered $\sigma_i$) | 42.61 (0.99) | 48.48 (0.98) | **81.78** (1.00) |
| + Llama SVF (shuffled $\sigma_i$) | 41.93 (0.98) | 46.34 (0.94) | 80.81 (0.99) |

### B.4 TRAINING CURVE OF LORA AND POLICY GRADIENT

Figure 9 gives the learning curves for LoRA training on the GSM8K task.

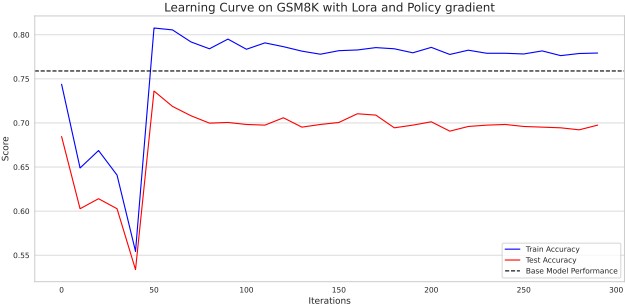

Figure 9: **Training LoRA with policy gradient.** The dashed line shows the performance of LLAMA3-8B-INSTRUCT on the test split. LoRA collapses at the beginning of the training stage and fails to recover, leading to negative effects on test performance. We swept a wide range of learning rates $(2 \times 10^{-4}, 5 \times 10^{-4}, \ldots, 2 \times 10^{-2}, 5 \times 10^{-2})$, and all learning curves were similar to the one presented.

## C    PCA ON LLAMA3 AND MISTRAL

To investigate if the singular components that have the highest singular values are able to capture most of the information of a weight matrix, we conducted Principle Component Analysis (PCA) on the weight matrices in LLAMA3-8B-INSTRUCT and MISTRAL-7B-INSTRUCT-V0.3 (see Figures 10 and 11). In each figure, we plot the variance that is captured by the top $r$ components across all the layers in each type of modules for a weight matrix $W \in \mathbb{R}^{m \times n}$:

$$\text{ratio} = \frac{\sum_{i=1}^{r} \sigma_i}{\sum_{j=1}^{\min(m,n)} \sigma_j}$$

Here, $\sigma$'s are the ordered (from largest to smallest) singular values on the weight matrix $W$. It is easy to see from the figures that when $r = 256$, less than 50% of the variance is captured by these top components on average. For the MLP layers, this fraction is even lower than 20%. On the other hand, the ranks adopted by LoRA-XS or similar methods are much less than 256, resulting in even more information loss and restrictions in their modeling power that relies mostly on these $r$ components.

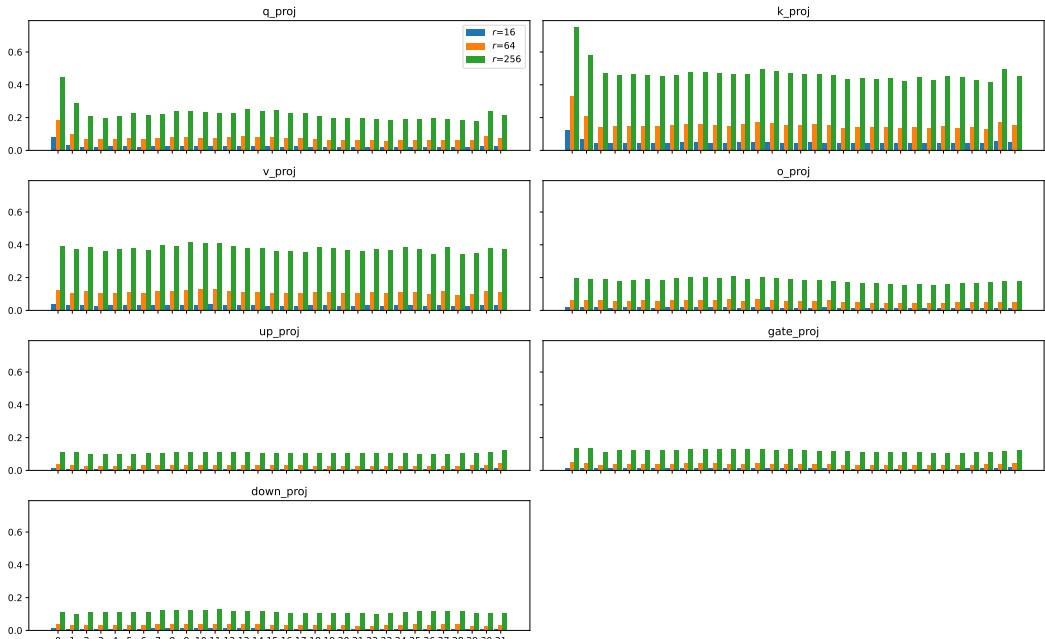

Figure 10: **PCA of LLAMA3-8B-INSTRUCT.** We show the ratio of the variance captured by the top $r$ singular components on the y-axis, and the layer indices on the x-axis. Except for the Query, Key and Value projection matrices, small $r$ values only capture a tiny fraction of variance in singular values in the parameter matrices.

## D    EFFICIENCY CONSIDERATIONS AND IMPROVEMENTS

Our CEM-based adaptation method involves running inference on a small number of samples for each target task (up to 10 in our experiments). In a typical configuration, this process is relatively efficient: for example, our CEM-light approach (3-shot with 10 generations) completes the ARC-Challenge task in approximately 11 minutes. As shown in Table 10, this lighter setup reduces the total number of samples to just 3% of the original setting while still delivering substantial performance improvements over the base model.

Table 10: **3-shot and light variants** Performance with different inference-time adaptation budgets.

| Method | ARC-Challenge |
|---|---|
| LLAMA3-8B-INSTRUCT | 80.63 (1.00) |
| + CEM 10-shot adaptation | **82.61 (1.02)** |
| + CEM 3-shot (30% of prompts) | 82.18 (1.02) |
| + CEM light (**3% of prompts**) | 82.08 (1.02) |

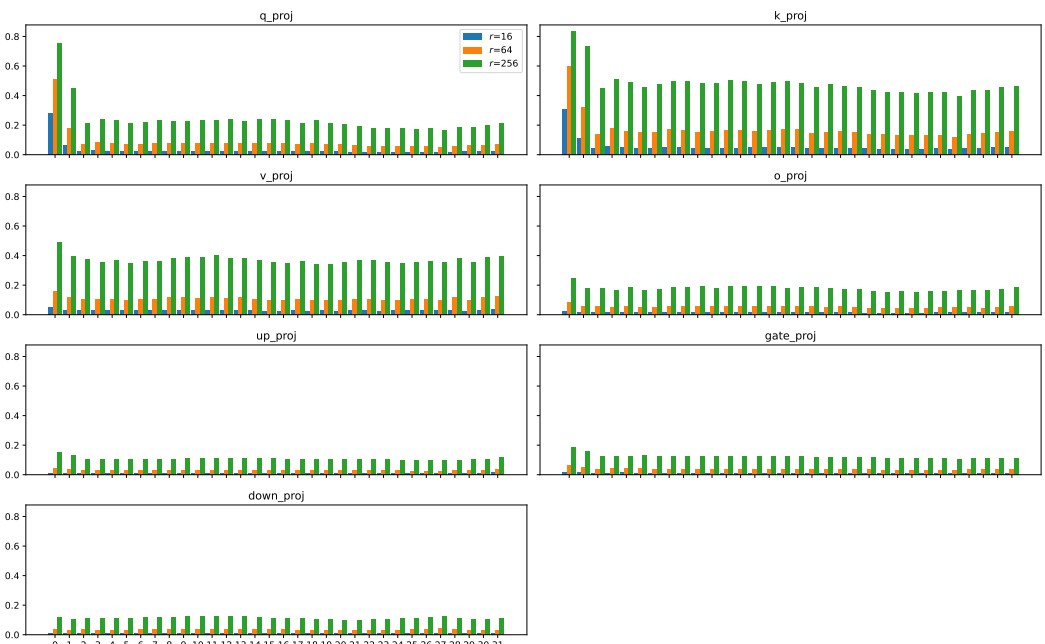

Figure 11: **PCA of MISTRAL-7B-INSTRUCT-V0.3.** We show the ratio of the variance captured by the top $r$ singular components on the y-axis, and the layer indices on the x-axis. Except for the Query, Key and Value projection matrices, small $r$ values only capture a tiny fraction of variance in singular values in the parameter matrices.

We acknowledge that CEM-based adaptation entails a trade-off between one-time overhead it spends on searching the optimal combination weights for the SVF-tune vectors and performance. Increasing the number of few-shot samples or the number of generations can yield higher performance, but this comes at the cost of additional computational overhead. However, it is important to note that this adaptation cost is a one-time overhead per task. The cost-per-prompt diminishes significantly when applied to tasks with a large number of prompts.

Moreover, in practical scenarios, CEM-based adaptation offers better scalability than few-shot prompting methods, which require increasing the length of every prompt, leading to much worse scaling as task sizes grow. In contrast, our method focuses on determining optimal expert vector combinations efficiently and avoids repetitive inference-time costs. However, we note that the overhead might be significant for tasks with very few prompts. Thus, the other adaptations methods might be more appropriate for these particular settings.

We also highlight two immediate directions for improving efficiency:

1. Reducing the number of few-shot samples: As shown in our ablation study in Appendix B.2, substantial benefits can be seen even in the 3-shot setting, which requires only evaluation of only 30% of the number of prompts per generation.

2. Reducing the number of maximum generations: In the explored settings, the CEM parameters tend to converge early on, being very close to the final values after a much lower number of generations than 100.

Finally, in this work we only considered CEM due to its simplicity, there exist several different evolution algorithms empirically showing better efficiency and convergence properties that we hope will be explored in future research.

