# OpenReview forum: "Transformer-Squared: Self-adaptive LLMs"
_ICLR.cc/2025/Conference — ICLR 2025 Poster_

### Official Review · Reviewer_Kvfw · 2024-10-29

**Soundness:** 2
**Presentation:** 3
**Contribution:** 2
**Rating:** 6
**Confidence:** 5

**Summary:**

This work proposes a framework called $\text{Transformer}^2$ to enable self-adaptation in LLMs.
In the initial stage of $\text{Transformer}^2$, task-specific components of the pre-trained weights are determined by leveraging SVF,  a noverl parameter-efficient fine-tuning approach.
SVF is based on learning weights for singular values after SVD decomposition on pre-trained weights.
In the second stage, $\text{Transformer}^2$ performs test-time adaptation by leveraging a few samples of test data to infer how to select or mix the task-specific components.

**Strengths:**

- For the most part the paper is easy to follow and well written.

- SVF reduces the parameter-count compared to standard LoRA.

- $\text{Transformer}^2$ can adapt quickly to the holdout test tasks.

- Compelling results for cross-model compatibility of task-specific components.

**Weaknesses:**

- **Weak and missing baseline results**

Most results for LoRA in Table 1 show worse results than the base model, how is this possible? In [1], for example, LoRA even outperforms full fine-tuning on MBPP. A performance drop after fine-tuning indicates that something went wrong with LoRA fine-tuning, maybe suboptimal hyperparameters?  [2] demonstrates importance of learning rate (higher is often better for LoRA), [3] demonstrates sensitivity w.r.t. scaling parameter, maybe there is an issue there? Also, there is no additional info given on the rank used for LoRA and to what weight matrices LoRA is applied to, smaller rank may perform better on smaller datasets.
There have also been advances for LoRA in terms of initialization [1,4,5] that have shown to improve upon LoRA and should be considered.

Another missing baseline is IA3 [6] which is conceptually very similar to SVF as it inserts newly trainable scaling vectors in the model.
Especially in scarce data regime, IA3 has shown improvements over LoRA, therefore it should be included in the paper.
Finally, the MoE approaches are similar to SVF and there are extensions for LoRA [7], but those are not compared to either.

- **Fair Comparisons**

There is no fair baseline comparison for $\text{Transformer}^2$, as it leverages test samples.
For example, $\text{Transformer}^2$ (Cls-expert) and $\text{Transformer}^2$ (Prompt) leverage two forward passes, one to determine the class and another one for generation. Can the authors clarify if this additional forward pass is required for each test sample?

 $\text{Transformer}^2$ (Few-shot) even leverages few-shot exemplars of the test set for determining the weighting factors $\boldsymbol{\alpha}_k$.
Therefore a fair comparison would require the baseline to use an equal amount of compute as the $\text{Transformer}^2$ variants during inference.
To this end, I would suggest fine-tuning the pre-trained IA3 vectors on the few-shot exemplars.

Another missing comparison is to exchange SVF for the second stage with another PEFT method, e.g. LoRA, or IA3, that were trained via SFT.
This would give important insights no the necessity of SVF in combination with RL as PEFT method for the first stage.

- **Limitations**

The authors claim that $\text{Transformer}^2$ enables self-adaptive LLMs, even though a lot of components are handcrafted in their framework, i.e. reward function, number of pre-defined classes, training on a number of datasets associated with pre-defined classes, determining of weighting factors by repeated sampling. This limits applicability of the proposed framework and also questions to what extent $\text{Transformer}^2$ is truly self-adaptive. Therefore, I would suggest to caveat the claims on self-adaptiveness and discuss the mentioned limitations.

$\text{Transformer}^2$ is limited to information the LLM has obtained during pre-training, i.e. it will not work on tasks that have not been observed during pre-training. While the authors have mentioned in the conclusion that model-merging techniques may be used to overcome this issue, this is impracticable as it requires searching and applying SVF on other pre-trained models until a suitable task-specific component is found. This is a fundamental limitation of $\text{Transformer}^2$ which makes it less practical.
Another limitation that should be made more explicit is the dependence on architecture for cross-model experiments - if architecture differs this is not possible anymore.

* **Overclaiming:**
  * Claims on overfitting:
    * line 77: "our approach mitigates the risk of overfitting" - figure 4 learning curves show otherwise
    * line 339: "highlighting the particular effectiveness of our proposed fine-tuning recipe in narrow spe-
cialized applications." - the smallest dataset is MBPP where SVF only performs better on Mistral
  * line 332: "SVF effectively fine-tunes to surpass the base performance." - figure 4 curves on coding and reasoning show otherwise
  * line 350: "all of our $\text{Transformer}^2$ adaptation strategies demonstrate improvements across all tasks for both LLAMA 3-8B-INSTRUCT and LLAMA 3-70B-INSTRUCT base models" - not true for LLAMA 3-70B-INSTRUCT on Math
  * line 355: "LoRA’s parameterization and optimization might be particularly sensitive to overfitting, especially when trained with the smaller GSM8K and MBPP-Pro datasets" - GSM8K is an order of magnitude larger than MBPP
  * line 364: "$\text{Transformer}^2$ with few-shot self-adaptation is always the highest-scoring method, providing notable improvements across all tested settings" - not true for LLAMA 3-70B-INSTRUCT on Math
  * line 408: "$\text{Transformer}^2$ could provide foundation models with new means to continually improve performance when deployed in lifelong settings." - not true if new task is not represented in the base models' weights
  * line 418: "Furthermore, they also show that using the classification expert consistently provides higher classification accuracy than vanilla prompt engineering." - not shown for LLAMA 3-70B-INSTRUCT
  * line 476: "we first proposed SVF, offering superior performance than prior fine-tuning recipes, together with reduced costs, high compositionality, and overfitting regularization" - there is evidence in figure 4 for overfitting

I recommend rephrasing the above examples which, as far as I can see, are cases of overclaiming.

**References:**

[1] PiSSA: Principal Singular Values and Singular Vectors Adaptation of Large Language Models, Meng et al., NeurIPS 2024

[2] LoRA+: Efficient Low Rank Adaptation of Large Models, Hayou et al., ICML 2024

[3] A Rank Stabilization Scaling Factor for Fine-Tuning with LoRA, Kalajdzievski, arXiv 2023

[4] LoRA-GA: Low-Rank Adaptation with Gradient Approximation, Wang et al., arXiv 2024

[5] One Initialization to Rule them All: Fine-tuning via Explained Variance Adaptation, Paischer et al., arXiv 2024

[6] Few-Shot Parameter-Efficient Fine-Tuning is Better and Cheaper than In-Context Learning, Liu et al., NeurIPS 2022

[7] Mixture-of-LoRAs: An Efficient Multitask Tuning for Large Language Models, Feng et al., COLING 2024

**Questions:**

- Why is $\text{Transformer}^2$ bound to SVF as PEFT method, can it not be used with other ones, e.g. LoRA, IA3, etc.?
- Figure 7 shows approximately uniform weighting of components works well for Llama-3, how does uniform weighting work in general? This would be an insightful ablation study.
- line 252: "we find the regularization properties of SVF avoid many of the failure modes of prior less-constrained parameterizations" - What failure modes of RL in particular does SVF remedy?
- line 253: "Thus, combining these complementary components effectively enables us to avoid relying on expensive fine-tuning procedures with large hand-designed datasets as proxies, and directly maximize task performance end-to-end." - Why is this only a benefit of SVF, as it can be applied to LoRA as well?
- How expensive is the reward labelling prior to RL finetuning?

---

> ### Author Response · Authors · 2024-11-24
> **Response to Reviewer Kvfw - Part 1**
>
> We thank the reviewer for taking the time to provide thoughtful comments and suggestions. We are happy to see that the reviewer thinks our paper is well-written and that our results are compelling. The following are our responses to the comments and questions, please see the details in the revised PDF.
>
>
> > Most results for LoRA in Table 1 show worse results than the base model, how is this possible? In [1], for example, LoRA even outperforms full fine-tuning on MBPP. A performance drop after fine-tuning indicates that something went wrong with LoRA fine-tuning, maybe suboptimal hyperparameters? [2] demonstrates importance of learning rate (higher is often better for LoRA), [3] demonstrates sensitivity w.r.t. scaling parameter, maybe there is an issue there?
>
> We used the most commonly adopted hyperparameters for this experiment (they worked well for us in our past experiments) and were surprised to see the results as well. On the other hand, the data for LoRA training is smaller than our usual practice, and we therefore think this reflects LoRA’s sensitivity to the amount of data. Following the reviewer’s suggestion, we have conducted a hyperparameter sweep for LoRA, with an extra emphasis on tuning the learning rate. Please see summarized results in the table below, detailed results are in Tables 1 and 2 in the revised PDF.
>
> | Method                  | GSM8K    | MBPP-Pro | ARC-Easy | MATH     | Humaneval | ARC-Challenge |
> |-------------------------|----------|----------|----------|----------|-----------|---------------|
> | LLAMA3-8B-Instruct  | 75.89    | 64.65    | 88.59    | 24.54    | 60.98     | 80.63         |
> | + LoRA                  | 77.18    | **67.68**| 88.97    | 24.12    | 52.44     | 81.06         |
> | + IA$^3$                | 78.01    | **67.68**| 89.10    | 23.64    | 59.76     | 81.57         |
> | + DoRA                  | 78.09    | 64.65    | 89.14    | 24.44    | 52.44     | 81.14         |
> | + Ours                  | **79.15**| 66.67    | **89.56**| **25.47**| **62.99** | **82.61**     |
>
> > Also, there is no additional info given on the rank used for LoRA and to what weight matrices LoRA is applied to, smaller rank may perform better on smaller datasets. There have also been advances for LoRA in terms of initialization [1,4,5] that have shown to improve upon LoRA and should be considered.
>
> Please kindly refer to this information in Tables 4 and 5 in the appendix in the original draft, where in the main text, we also refer a reader to the Appendix for all the hyper-parameters and settings we used on lines 323 and 343.
>
> > Another missing baseline is IA3 [6] which is conceptually very similar to SVF as it inserts newly trainable scaling vectors in the model. Especially in scarce data regime, IA3 has shown improvements over LoRA, therefore it should be included in the paper. Finally, the MoE approaches are similar to SVF and there are extensions for LoRA [7], but those are not compared to either.
>
> We thank the reviewer for the suggestion and have included DoRA (suggested by Reviewer FCnJ) and IA$^3$  as additional baselines. While we tried our best to follow the reviewer's instructions, due to the limited rebuttal time and computing resources we have, these comparisons are for Llama3-8B-Instruct. We will add complete comparisons in the camera-ready version of the paper. The table above summarizes the new results, these new baselines are also in Table 7 in the revised PDF. Although all the baselines are able to outperform the Llama3-8B-Instruct model after hyperparameter sweeps, our method’s performance is still the best in general with the least trainable parameters.
>
> On the other hand, we don’t think MoE/MoA approaches are similar to SVF. They require the training of completely new experts and/or routers. Moreover, they are not as scalable as our methods with the increase of tasks (e.g., adding new E/As requires adjusting the router that may risk affecting the performance of the entire mixture, whereas in our method, we can simply tune the prompt). For these reasons, we did not add them as baselines. We are happy to discuss this further.

---

> > ### Author Response · Authors · 2024-11-24
> > **Response to Reviewer Kvfw - Part 2**
> >
> > > There is no fair baseline comparison for Transformer$^2$, as it leverages test samples. For example, Transformer$^2$ (Cls-expert) and Transformer$^2$ (Prompt) leverage two forward passes, one to determine the class and another one for generation. Can the authors clarify if this additional forward pass is required for each test sample?
> >
> > Yes, the additional forward pass is required for each test sample. However, this extra pass merely helps our method to adapt its internal weights and carries no new external information.
> >
> > > Transformer$^2$ (Few-shot) even leverages few-shot exemplars of the test set for determining the weighting factors $\alpha_k$. Therefore a fair comparison would require the baseline to use an equal amount of compute as the Transformer$^2$ variants during inference. To this end, I would suggest fine-tuning the pre-trained IA3 vectors on the few-shot exemplars.
> >
> > We appreciate and understand the reviewer’s concerns about fairness in our evaluation with CEM. Thus, following their suggestion, we have added results fine-tuning the full parameter set of IA$^3$ on the few-shot test samples in the ARC-Chellenge task. We added these results to Table 8 in Section B.2, and we summarize them in the table below for easier reference.
> >
> > | Number of shots | CEM-based adaption | IA$^3$ (100 steps) | IA$^3$ (1000 steps) |
> > | :---- | :---- | :---- |:----|
> > | 3 | 82.18 | 81.82 | 79.01|
> > | 5 | 82.38 | 80.89 | 79.41|
> > | 10 | 82.61 | 82.00 | 79.78|
> > | 20 | 82.61 | 81.40 | 79.61 |
> >
> > Our results indicate that the performance of IA$^3$ on the unseen test tasks is inferior to CEM-based adaptation for all numbers of few shots considered.
> >
> > We note that in our experiment, we have to considerably limit the number of optimization steps to avoid overfitting the 500,000 parameters of IA$^3$ on the few-shot samples. However, we believe overfitting might still be occurring to some degree even after only 100 steps, as also validated by the model’s perfect training accuracy on this extremely small dataset.
> >
> > In contrast, we also note that our CEM adaptation mechanism only optimizes for a number of parameters (the $\alpha_k$) equal to the number of the expert vectors (only 3 in our case). These properties, together with the analyzed inherent composability and expressiveness of our SVF pre-training even show we can obtain substantial performance benefits superior to IA$^3$ with as few as 3 few-shot samples.
> >
> > We hope these newly added experiments and analysis better highlight the current limitations of using direct finetuning on an extremely small number of data points, together with the potential advantages of self-adaption with few-shot samples.
> >
> > > Another missing comparison is to exchange SVF for the second stage with another PEFT method, e.g. LoRA, or IA3, that were trained via SFT. This would give important insights no the necessity of SVF in combination with RL as PEFT method for the first stage.
> >
> > We agree with the reviewer that the LLM weights for the 2nd pass can be replaced with those trained from other PEFT methods. However, this is not a missing comparison since we are not claiming the general applicability of the 1st inference pass. The advantage of SVF lies in its superior performance brought by extremely few trainable parameters. This has been proved by Table 1 in the original draft and the new results we presented above. We are happy to further discuss this with the reviewer.
> >
> >
> > > The authors claim that Transformer$^2$ enables self-adaptive LLMs, even though a lot of components are handcrafted in their framework, i.e. reward function, number of pre-defined classes, training on a number of datasets associated with pre-defined classes, determining of weighting factors by repeated sampling. This limits applicability of the proposed framework and also questions to what extent Transformer$^2$ is truly self-adaptive. Therefore, I would suggest to caveat the claims on self-adaptiveness and discuss the mentioned limitations.
> >
> > With all respect, we find this comment confusing. The handcrafted components are engineering design choices, and we don’t see conflicts to the resulting system being able to self-adapt to the test examples. Concretely,
> > - A reward function is necessary for any RL to work, and we don’t have a complex design.
> > - Number of pre-defined classes is the number of skills, a trivial yet reasonable parameter for self-adaptation.
> > - Training on associated datasets is necessary for any ML systems.
> > - Determining the weighting factors is how our method self-adapts, this process is automatic, without manual inference or any help from external systems. Furthermore, there is no repeated sampling involved in our system.
> >
> > We tried our best to provide answers and would appreciate the reviewer’s feedback on how we can modify our draft to eliminate these misunderstandings and address the reviewer’s concern.

---

> > > ### Author Response · Authors · 2024-11-24
> > > **Response to Reviewer Kvfw - Part 3**
> > >
> > > > Transformer$^2$ is limited to information the LLM has obtained during pre-training, i.e. it will not work on tasks that have not been observed during pre-training. While the authors have mentioned in the conclusion that model-merging techniques may be used to overcome this issue, this is impracticable as it requires searching and applying SVF on other pre-trained models until a suitable task-specific component is found. This is a fundamental limitation of Transformer$^2$ which makes it less practical.
> > >
> > > We think there is misunderstanding about what we meant by leveraging model merging. Specifically, there is no need to search and apply SVF on other pretrained models, we apologize if our writing in the original draft led the reviewer to this misunderstanding.
> > >
> > > As the reviewer pointed out (we also mentioned in the conclusion), the performance of Transformer$^2$ is limited to the capability of the base model (this applies to all PEFT trained models too). While there are many ways to enhance the base model, we see the model merging-based approach promising, because in contrast to fine-tune an LLM to be good at multiple skills at once, model merging-based approach allows one to train a set of LLMs to each develop separate skills and merge the set to acquire one generally improved LLM.
> > >
> > > One can follow the model merging recipe and create a better model out of seed models of known capabilities. Furthermore, we also observe merged models starting to dominate open leaderboards whose performance on well-known tasks/capabilities are reported. These models also make natural candidate base models for Transformer$^2$ to build on.
> > >
> > > We have modified the text in the conclusion (see Sec 5) to clarify this point. We would also appreciate it if the reviewer could suggest modifications to the draft to clear the misunderstanding.
> > >
> > > > Another limitation that should be made more explicit is the dependence on architecture for cross-model experiments - if architecture differs this is not possible anymore.
> > >
> > > We agree that cross-model experiments cannot be carried out if the model architectures are drastically different. On the other hand, cross-model transfer is a surprising finding that we wish to share with the community. We stated clearly on line 458 in the original draft that it is an open research question, it is not what we are claiming in the paper and is therefore not a limitation of the work.

---

> ### Author Response · Authors · 2024-11-24
> **Response to Reviewer Kvfw - Part 4**
>
> > line 77: "our approach mitigates the risk of overfitting" - figure 4 learning curves show otherwise
>
> We apologize for the confusion in Figure 4, we have updated it and its caption for clarification. In short, while we used the best validation score to select our checkpoint for evaluation (marked by red dots) in the figure, we present longer training curves without early stopping to demonstrate \svdacro's learning capabilities. Tasks with only hundreds of training samples like Coding and Reasoning were stopped early (after 10 and 80 epochs). We update the parameters at the end of each epoch.
>
> Generally speaking, for overparameterized ML systems without special treatment to prevent overfitting (e.g., dropout, regularization), the gap between training and test learning curves will decrease at first and then depart as the training proceeds. The reviewer’s statement about Figure 4 should not be an argument for our claim on line 77. Instead, our claim is based on the results in Tables 1, 2 and the new results reported above, where SVF performs better on the test split and the transfer tasks than the baselines most of the time.
>
> > line 339: "highlighting the particular effectiveness of our proposed fine-tuning recipe in narrow spe- cialized applications." - the smallest dataset is MBPP where SVF only performs better on Mistral
>
> We have removed this sentence in the revision.
>
> > line 332: "SVF effectively fine-tunes to surpass the base performance." - figure 4 curves on coding and reasoning show otherwise
>
> As stated previously, we picked the snapshots for evaluation based on the best validation performance. We have modified Figure 4 in the revision to highlight these points. All of them are above the baseline curve.
>
> > line 350: "all of our Transformer$^2$ adaptation strategies demonstrate improvements across all tasks for both LLAMA 3-8B-INSTRUCT and LLAMA 3-70B-INSTRUCT base models" - not true for LLAMA 3-70B-INSTRUCT on Math
>
> We have modified the sentence to exclude Llama3-70B-Instruct on the Math task.
>
> > line 355: "LoRA’s parameterization and optimization might be particularly sensitive to overfitting, especially when trained with the smaller GSM8K and MBPP-Pro datasets" - GSM8K is an order of magnitude larger than MBPP
>
> We are not claiming anything in this sentence and are not comparing the data sizes of GSM8K and MBPP-Pro. Can the reviewer clarify why they think this is an overclaim?
>
> > line 364: "Transformer$^2$ with few-shot self-adaptation is always the highest-scoring method, providing notable improvements across all tested settings" - not true for LLAMA 3-70B-INSTRUCT on Math
>
> We have modified the sentence to exclude Llama3-70B-Instruct on the Math task.
>
> > line 408: "Transformer$^2$ could provide foundation models with new means to continually improve performance when deployed in lifelong settings." - not true if new task is not represented in the base models' weights
>
> While the reviewer is correct that it would be challenging if the desired skill is not embedded in the base model, our sentence is more general and includes a lot of untapped skills that already exist in the base model. Besides, we have pointed out this limitation in the conclusion, which applies to all PEFT methods.
>
> > line 418: "Furthermore, they also show that using the classification expert consistently provides higher classification accuracy than vanilla prompt engineering." - not shown for LLAMA 3-70B-INSTRUCT
>
> We have modified the text to exclude Llama3-70B-Instruct.
>
> > line 476: "we first proposed SVF, offering superior performance than prior fine-tuning recipes, together with reduced costs, high compositionality, and overfitting regularization" - there is evidence in figure 4 for overfitting
>
> As we explained earlier, Figure 4 should not be used as an argument for overfitting. Our results in Tables 1, 2 and the new results reported above support our claim.

---

> > ### Author Response · Authors · 2024-11-24
> > **Response to Reviewer Kvfw - Part 5**
> >
> > > Why is Transformer$^2$ bound to SVF as PEFT method, can it not be used with other ones, e.g. LoRA, IA3, etc.?
> >
> > As we discussed earlier, it technically can adopt PEFT methods in the 2nd pass. However, the performance we observed in the experiments and the extremely few extra parameters makes SVF the most appropriate method for self-adaptation.
> >
> > > Figure 7 shows approximately uniform weighting of components works well for Llama-3, how does uniform weighting work in general? This would be an insightful ablation study.
> >
> > In Figure 7, only the top right diagram (Mistral@MATH) shows approximately uniform weighting, Llama-3’s results are not uniform. We kindly ask the reviewer to confirm.
> >
> > > line 252: "we find the regularization properties of SVF avoid many of the failure modes of prior less-constrained parameterizations" - What failure modes of RL in particular does SVF remedy?
> >
> > We observe few oscillations and instabilities in the training performance, which are common in RL training and tend to require very low learning to avoid performance collapse (e.g. as commonly seen in RLHF and even for our LoRA baselined trained with RL). We conjecture this is mostly due to the parameterization scheme SVF.
> >
> > > line 253: "Thus, combining these complementary components effectively enables us to avoid relying on expensive fine-tuning procedures with large hand-designed datasets as proxies, and directly maximize task performance end-to-end." - Why is this only a benefit of SVF, as it can be applied to LoRA as well?
> >
> > LoRA fine-tuning requires labeled datasets for next token prediction (what we meant by “hand-designed datasets”). Trained with RL, its result is not as good as SVF. These results are shown in Tables 1, 2 and the new results above.
> >
> > > How expensive is the reward labelling prior to RL finetuning?
> >
> > We don’t label rewards prior to RL, the reward design is simple and explained on line 244. In implementation, it only costs a single inference pass.

---

> > > ### Comment · Reviewer_Kvfw · 2024-11-25
> > >
> > > I greatly appreciate the clarifications and the additional results!
> > > There are still a few remaining questions from my side:
> > >
> > > **Baselines:**
> > >
> > > Thank you for adding results on IA3 and adding additional experiments which verified my suspicion that LoRA was not well tuned.
> > > To clarify, am I correct with the assumption that all results in Table 1 for LoRA stem from next-token prediction?
> > > I believe this is an interesting result that SVF seems to work really well only in combination with RL, whereas LoRA only works well with SFT.
> > >
> > > With MoE I was actually referring to a baseline for Transformer², not for SVF. This is mostly because the authors classify both approaches as microview on internal LLM adaptation. Therefore I was expecting it to be compared to as a baslines for Transformer².
> > >
> > > **Additional complexity:**
> > >
> > > Firstly, I would like to apologize for my comment on self-adaptiveness, which have caused some confusion.
> > > My point was that there are many handcrafted components that incur additional cost. In particular:
> > > - For SVF we need to label a dataset with rewards, for standard LoRA fine-tuning we do not, we simply frame the task as SFT. RL is usually more expensive than SFT, especially for sparse rewards (line 244, one scalar reward for a whole answer).
> > > - The definition of skill classes requires human prior, i.e. it is defined by humans based on what would be reasonable choices given the pre-training data. Therefore my argument, the ability for "self-adaptiveness" is not given.
> > > - As far as I understood, Transformer²(Prompt) and Transformer²(Cls-expert) require one additional forward pass, Table 3 shows that it depends very much on the downstream task how much cost it incurs. For ARC-challenge it doubles the cost as the final task only requires a few token generation. For MATH it incurs less cost, but the claim that it is negligible is not supported.
> > > - I am not entirely certain on this point, but Transformer²(Few-shot) requires more than one forward pass because it leverages CEM to find values for $\alpha_k$, right? If so, how many forward passes are these? The improvement from Transformer²(Cls-expert) to Transformer²(Few-shot) is very little, does it justify leveraging a lot more forward passes?
> > >
> > > **Minor confusions:**
> > >
> > > > As the reviewer pointed out (we also mentioned in the conclusion), the performance of Transformer
> > >  is limited to the capability of the base model (this applies to all PEFT trained models too).
> > >
> > > I disagree here, LoRA introduces new trainable parameters into the model which provide additional complexity to store more knowledge. SVF only scales singular values therefore it is less expressive and cannot store new knowledge.
> > >
> > > > In Figure 7, only the top right diagram (Mistral@MATH) shows approximately uniform weighting, Llama-3’s results are not uniform. We kindly ask the reviewer to confirm.
> > >
> > > Sorry, I meant Llama3-8B@HumanEval are approximately uniform, as well as Mistral7B@MATH.
> > > If setting alpha values uniform already performs very well, then additional cost might be saved by simply doing so.
> > >
> > > > line 355: "LoRA’s parameterization and optimization might be particularly sensitive to overfitting, especially when trained with the smaller GSM8K and MBPP-Pro datasets" - GSM8K is an order of magnitude larger than MBPP
> > >
> > > Sorry about the confusion, my point here was that GSM8K is not one of the smaller datasets. It is actually the largest one that have been used as far as I can see.
> > >
> > > The remainder of my points have been properly addressed. Since I find that the part on PEFT via RL is interesting I would be willing to increase my score given that claims on "negligible increase in overall inference time" will be corrected.

---

> > > > ### Author Response · Authors · 2024-11-26
> > > > **Further Response to Reviewer Kvfw - Part 1**
> > > >
> > > > We thank the reviewer for their active engagement in the discussion and for clarifying their original feedback.
> > > >
> > > > > Thank you for adding results on IA3 and adding additional experiments which verified my suspicion that LoRA was not well tuned. To clarify, am I correct with the assumption that all results in Table 1 for LoRA stem from next-token prediction? I believe this is an interesting result that SVF seems to work really well only in combination with RL, whereas LoRA only works well with SFT.
> > > >
> > > > Yes, all LoRA results in Table 1 are from next-token predictions. We also find it interesting, and have added a sentence on line 351 to comment on this.
> > > >
> > > > > With MoE I was actually referring to a baseline for Transformer², not for SVF. This is mostly because the authors classify both approaches as microview on internal LLM adaptation. Therefore I was expecting it to be compared to as a baslines for Transformer².
> > > >
> > > > We thank the reviewer for the clarification. We will be sure to collect and add results also for this additional experiment to the camera-ready revision.
> > > >
> > > > > 1. For SVF we need to label a dataset with rewards, for standard LoRA fine-tuning we do not, we simply frame the task as SFT. RL is usually more expensive than SFT, especially for sparse rewards (line 244, one scalar reward for a whole answer).
> > > >
> > > > Thanks for this clarification and further discussion. Works leveraging RLHF indeed need human-annotated rewards and are more costly. However, this is not the case in our method, as we adopt a simple right-or-wrong automated comparison between a generated answer and the ground-truth to give a reward (i.e., a binary classification). Furthermore, for the considered set of tasks, we also do not seem to incur the efficiency problems related to sparse rewards that the reviewer is describing.
> > > >
> > > > However, we understand the reviewer’s concerns that manual human-annotations and sparse rewards could be issues when applying our methods to a wider range of task. Thus, we have added text on line 257 to discuss this, which we agree could be relevant for future extensions of our work.
> > > >
> > > >
> > > > > 2. The definition of skill classes requires human prior, i.e. it is defined by humans based on what would be reasonable choices given the pre-training data. Therefore my argument, the ability for "self-adaptiveness" is not given.
> > > >
> > > > Thank you for the clarification, we have a better vision of where this misunderstanding is rooted from. We think the mapping between a dataset and its corresponding skill class (e.g., GSM8K and MATH belong to math, MBPP and Humaneval belong to coding) is easy to establish manually. In fact, these categorizations are almost always given in the dataset's meta data. However, we understand that these relationships might not be always exact. Following the reviewer's suggestion, we added a sentence on line 267 for clarification.
> > > >
> > > > > 3. As far as I understood, Transformer²(Prompt) and Transformer² (Cls-expert) require one additional forward pass, Table 3 shows that it depends very much on the downstream task how much cost it incurs. For ARC-challenge it doubles the cost as the final task only requires a few token generation. For MATH it incurs less cost, but the claim that it is negligible is not supported.
> > > >
> > > > We appreciate the opportunity for us to further clarify this point. We understand the reviewer's point that saying "neglibile" is not well supported. Thus, we modified this statement and quantified these extra costs (see line 418 and Table 3):
> > > >
> > > > *... In our settings, it is $\mathcal{O}(n)$ where $n$ is the length of the input. ARC-challenge's cost ratio is large because they are single choice problems and therefore the cost of the 2nd pass is also $\mathcal{O}(n)$. In general settings, we think it is reasonable to assume this ratio to be closer to those of MATH and Humaneval. Transformer$^2$ spends small extra time in exchange for improved performance and the capability for self-adaptation.*
> > > >
> > > > We are open to tune this claim further if the reviewer still has any concerns.

---

> > > > ### Author Response · Authors · 2024-11-26
> > > > **Further Response to Reviewer Kvfw - Part 2**
> > > >
> > > > > 4. I am not entirely certain on this point, but Transformer²(Few-shot) requires more than one forward pass because it leverages CEM to find values for, right? If so, how many forward passes are these? The improvement from Transformer²(Cls-expert) to Transformer²(Few-shot) is very little, does it justify leveraging a lot more forward passes?
> > > >
> > > > As the reviewer correctly points out, our Few-shot adaptation requires 32 batched forward passes per generation with the current hyper-parammeters. However, we wish to emphasize that this is a one-time cost that does not scale with the number of samples in each task (unlike traditional few-shot prompting and the other adaptation methods). Furthermore, we added Appendix D in the rebuttal revision, where we show even just 10 generations are enough for some tasks. Thus, while we agree this might be a significant bottleneck for tasks with few samples, we note the actual overhead might actually be smaller than other adaptation methods for tasks with many samples.
> > > >
> > > > Following the reviewer’s feedback, we expanded Section D to provide the above discussion and better acknowledge this tradeoff.
> > > >
> > > > >**Minor confusions:**
> > > > > 1. I disagree here, LoRA introduces new trainable parameters into the model which provide additional complexity to store more knowledge. SVF only scales singular values therefore it is less expressive and cannot store new knowledge.
> > > >
> > > > Thank you for bringing up the discussion about the knowledge storage. While we share the view that LoRA might offer better expressiveness due to its controllable rank and larger number of trainable parameters, we wish to point out that SVF learns a scaling factor for singular values that affects the weight matrix in a full rank manner. From this perspective, SVF is technically more informational than low rank approaches.
> > > >
> > > > We find the discussion from both sides helpful for readers to better understand the property of our approach, we have added discussion on line 227 to acknowledge this.
> > > >
> > > > > 2. Sorry, I meant Llama3-8B@HumanEval are approximately uniform, as well as Mistral7B@MATH. If setting alpha values uniform already performs very well, then additional cost might be saved by simply doing so.
> > > >
> > > > We appreciate your observation and suggestion. Indeed, for Llama3-8B@HumanEval and Mistral7B@MATH, approximately uniform alpha values yield strong results. This suggests an interesting possibility for efficiency in certain scenarios.
> > > >
> > > > However, our early experimental results across different tasks showed that uniform alpha values were not consistently effective. For instance, applying uniformly the three experts to LLAMA3-8B achieved 24.46 on MATH, while the original model achieves 24.54 and Transformer$^2$ (Few-shot) achieves 25.47. This suggests the importance and effectiveness of CEM adaptation and we currently lack reliable methods to determine apriori which scenarios will benefit from uniform aplhas.
> > > >
> > > > We have expanded our Analysis 2 to incorproate this discussion and better acknowledge the potential efficiency gains and limitations from uniform alpha values.
> > > >
> > > > > 3. Sorry about the confusion, my point here was that GSM8K is not one of the smaller datasets. It is actually the largest one that have been used as far as I can see.
> > > >
> > > > Thank you for the clarification.
> > > >
> > > > > The remainder of my points have been properly addressed. Since I find that the part on PEFT via RL is interesting I would be willing to increase my score given that claims on "negligible increase in overall inference time" will be corrected.
> > > >
> > > > We are pleased to hear that our previous responses have addressed most of the reviewer's concerns. We greatly appreciate the reviewer's thoughtful engagement and willingness to consider raising their score. We hope our answer to Point 3 above has satisfactorily clarified the claims on inference time. If so, we would be grateful for the reviewer's recommendation for acceptance, as we believe this paper presents several exciting findings that will contribute meaningfully to the community at ICLR.

---

> > > > > ### Comment · Reviewer_Kvfw · 2024-11-26
> > > > >
> > > > > Thank you for futher clarifying the raised points.
> > > > > I am addressing my remaining points as follows.
> > > > >
> > > > > > Furthermore, for the considered set of tasks, we also do not seem to incur the efficiency problems related to sparse rewards that the reviewer is describing.
> > > > >
> > > > > Well, this depends on the sparsity of the reward, a more dense reward signal would lead to even faster learning :)
> > > > >
> > > > > > However, we understand the reviewer’s concerns that manual human-annotations and sparse rewards could be issues when applying our methods to a wider range of task. Thus, we have added text on line 257 to discuss this, which we agree could be relevant for future extensions of our work.
> > > > >
> > > > > Thank you, my main point is that there IS human annotation involved even though it may be straightforward, it makes SVF less accessible than, e.g. plain SFT. For SFT you only need the token sequences, for SVF you additional need reward labeling first.
> > > > >
> > > > > > Thank you for the clarification, we have a better vision of where this misunderstanding is rooted from. We think the mapping between a dataset and its corresponding skill class (e.g., GSM8K and MATH belong to math, MBPP and Humaneval belong to coding) is easy to establish manually.
> > > > >
> > > > > I totally agree, it is easy for a human. Therefore my point that it is not straightforward to automatize, in turn my questioning of self-adaptiveness.
> > > > >
> > > > > > Transformer² spends small extra time in exchange for improved performance and the capability for self-adaptation.
> > > > >
> > > > > Can the authors remove this line? I believe the previous line puts it very well into perspective.
> > > > > Also, I presume Table 3 talks about Transfomer²(Cls-expert) or Transformer²(Prompt), this should be added.
> > > > >
> > > > > > As the reviewer correctly points out, our Few-shot adaptation requires 32 batched forward passes per generation with the current hyper-parammeters. However, we wish to emphasize that this is a one-time cost that does not scale with the number of samples in each task (unlike traditional few-shot prompting and the other adaptation methods). Furthermore, we added Appendix D in the rebuttal revision, where we show even just 10 generations are enough for some tasks.
> > > > >
> > > > > Thank you for clarifying, that makes sense. Could you add a reference from the text around Table 3 to the appendix D referring for discussion on efficiency of Transformer²(Few-shot)?
> > > > >
> > > > > > From this perspective, SVF is technically more informational than low rank approaches.
> > > > >
> > > > > This is a fair point regarding overall expressivity, however my main point was more on information that is not contained in the pre-trained models weights. SVF can only up/down-scale existing info, while LoRA can add new info. The authors have shown that this can work very well, my concern though is that this might be very limiting when considering downstream tasks that are more distinct.
> > > > >
> > > > > I will update as soon as the suggested changes are included in the manuscript.

---

> > > > > > ### Author Response · Authors · 2024-11-26
> > > > > > **Further Response to Reviewer Kvfw**
> > > > > >
> > > > > > As suggested, in the latest revision we now have:
> > > > > > 1. Removed the line “Transformer² spends small extra time in exchange for improved performance and the capability for self-adaptation.”
> > > > > > 2. Specified in the caption of of Table 3 and on line 410 that the results are about Transformer²(Prompt).
> > > > > > 3. Added a reference in the text around Table 3 (line 421) to Appendix D, referring readers to the discussion on efficiency of Transformer² (Few-shot).
> > > > > >
> > > > > > We would also like to take the opportunity to thank the reviewer once again for the clarifications and appreciate the efforts they have put in to provide us with concrete feedbacks. Please, do not hesitate to let us know if we are still missing anything.

---

> > > > > > > ### Comment · Reviewer_Kvfw · 2024-11-26
> > > > > > >
> > > > > > > Thank you for addressing the raised issues.
> > > > > > >
> > > > > > > I have raised my score since I believe the authors have brought forward valuable insights to the community.
> > > > > > > As mentioned in the previous discussion, I believe the paper can further benefit from additional baselines to Transformer² that leverage samples at test time.
> > > > > > > I am also thankful to the authors for putting the method in perspective in terms of efficiency, such that users can properly decide for themselves whether this is the correct approach for downstream adaptation.

---

### Official Review · Reviewer_cbgq · 2024-11-03

**Soundness:** 3
**Presentation:** 3
**Contribution:** 3
**Rating:** 6
**Confidence:** 3

**Summary:**

This paper proposes a self-adaptive framework called Transformer$^2$ that learns and adapts external expert modules to downstream tasks in real-time. These modules can be used to augment LLMs for specific abilities and provide more flexibility than traditional post-training approaches that directly update the LLM weights. At training time, unlike existing methods that learn the expert modules via LoRA, this paper proposes a method called SVF that tunes only the singular values of a weight matrix to reduce storage and computational costs. At test time, the Transformer$^2$ framework requires two forward passes, one to gather test-task-related signals to select and combine the expert vectors, and one to actually generate the answer with the adapted LLM. Experiments on multiple open-source LLMs and datasets show that the proposed framework outperforms existing self-adaptive methods with reduced adaptation costs.

**Strengths:**

1. The idea of tuning singular values is novel and beneficial for reducing storage and computational costs.
2. The idea of learning a dispatch module using the LLM itself that determines which expert modules should be used at adaptation time is interesting .
3. Extensive experiment results show the efficacy of the proposed methods.
4. Good presentation.

**Weaknesses:**

1. I think ablation studies in B.1 should be moved to main text because it provides a better understanding of how each component of Transformer$^2$ contributes to the performance gain. Also, is it possible to provide results for LoRA + next-token prediction in Table 5? Ideally with both attention and attention+MLP. Currently it's hard to conclude that SVF with policy gradient does not sacrifice fine-tuning performance.
2. Scaling to a large number of abilities or specialized domains might be an issue for the few-shot adaptation scheme that uses CEM to search?
3. How should we reconcile the test time performance on coding in Figure 4? The performance seems to be decreasing over time and worse than non-adapted model

Other minor issues:
1. Line 74: It's better to add the full name here because the abbreviation SVF appears for the first time.

**Questions:**

See weaknesses

---

> ### Author Response · Authors · 2024-11-24
> **Response to Reviewer cbgq**
>
> We appreciate the detailed review comments and concrete suggestions. In addition, we are happy to learn that the reviewer thinks our method to be novel, interesting and that our presentation is good. The following are our responses to the comments and questions, please see the details in the revised PDF.
>
> > I think ablation studies in B.1 should be moved to main text because it provides a better understanding of how each component of Transformer$^2$ contributes to the performance gain. Also, is it possible to provide results for LoRA + next-token prediction in Table 5? Ideally with both attention and attention+MLP. Currently it's hard to conclude that SVF with policy gradient does not sacrifice fine-tuning performance.
>
> We agree that moving the ablation study to the main text helps understanding the importance of each component better, and we have moved that part to the main text as Analysis 3 in Sec 4.3.
>
> Following the reviewer's suggestion, we have added LoRA + next-token-prediction in Table 4 in the revised PDF. This new setting’s performance is better than the base model, however it still underperforms our method, concluding that SVF is a more effective parameter-efficient training method.
>
> > Scaling to a large number of abilities or specialized domains might be an issue for the few-shot adaptation scheme that uses CEM to search?
>
> We thank the reviewer for pointing this out. As the number of abilities or specialized domains increases, we anticipate that the one-time overhead of our CEM-based adaptation (i.e., the time required to determine the optimal combination weights for SVF-tuned vectors) will also increase. We acknowledge this as a limitation and have explicitly discussed it in Section 5. However, we wish to emphasize that this is a one-time cost that does not scale with the number of samples in each task (unlike traditional few-shot prompting and the other adaptation methods). Furthermore, we believe this cost is offset by the benefits of enhanced self-adaptation capabilities and improved performance.
>
>
> > How should we reconcile the test time performance on coding in Figure 4? The performance seems to be decreasing over time and worse than non-adapted model
>
>
> We apologize for the confusion in Figure 4, we have updated it and its caption for clarification. In short, while we used the best validation score to select our checkpoint for evaluation (marked by red dots) in the figure, we present longer training curves without early stopping to demonstrate \svdacro's learning capabilities. Tasks with only hundreds of training samples like Coding and Reasoning were stopped early (after 10 and 80 epochs). We update the parameters at the end of each epoch.
>
> > Line 74: It's better to add the full name here because the abbreviation SVF appears for the first time.
>
> We thank the reviewer for the suggestion. We have added the full name before SVF appears for the first time.

---

> > ### Comment · Reviewer_cbgq · 2024-11-24
> >
> > I appreciate the authors' response. I'll keep my positive rating.

---

### Official Review · Reviewer_MTB2 · 2024-11-03

**Soundness:** 3
**Presentation:** 3
**Contribution:** 3
**Rating:** 6
**Confidence:** 4

**Summary:**

The paper presents Transformer2, a self-adaptive framework that enhances the adaptability of LLMs for unseen tasks in real-time. It introduces a parameter-efficient approach by adjusting only the singular components of their weight matrices. The inference is a two-pass mechanism which first identifies task properties, and then dynamically combines task-specific expert vectors. The expert vector is trained by reinforcement learning to generate task-specific response. Proposed method is shown to outperform existing parameter-efficient tuning methods like LoRA, demonstrating efficiency and versatility across different LLM architectures and multimodal tasks.

**Strengths:**

- Transformer2 introduces a parameter-efficient adaptation method that selectively tunes only singular components, reducing computational demands compared to full fine-tuning approaches.
- The use of a two-pass mechanism to dynamically adapt model responses enhances the real-time adaptability of LLMs for unseen tasks.
- Training expert vectors with reinforcement learning enables the model to tailor its performance, leading to improvements in task-specific outputs.
- The framework is adaptable to various LLM architectures and applicable to vision-language tasks, showcasing broad utility.

**Weaknesses:**

- While the paper proposes novel methods, the description of how these components interact and integrate in practice could be clearer.
- While the use of reinforcement learning to train expert vectors is an innovative aspect of the framework, it introduces significant computational overhead and complexity. This can potentially undermine the paper's claims of efficiency and raises the question of whether these additional costs were considered when comparing the method to others like LoRA. The experiments should include a detailed breakdown of this computational expense to provide a clearer picture of the method's practical applicability and efficiency.

**Questions:**

Refer to the weakness.

---

> ### Author Response · Authors · 2024-11-24
> **Response to Reviewer MTB2**
>
> We thank the reviewer for the feedback and concrete actionable suggestions. We are glad that the reviewer thinks our method to be novel and demonstrates broad utility. The following are our responses to the comments and questions, please see the details in the revised PDF.
>
> > While the paper proposes novel methods, the description of how these components interact and integrate in practice could be clearer.
>
> We have added text in “Methods” to clarify the interaction and integration of the components, please see line 262 in the revised PDF for details.
>
> > While the use of reinforcement learning to train expert vectors is an innovative aspect of the framework, it introduces significant computational overhead and complexity. This can potentially undermine the paper's claims of efficiency and raises the question of whether these additional costs were considered when comparing the method to others like LoRA. The experiments should include a detailed breakdown of this computational expense to provide a clearer picture of the method's practical applicability and efficiency.
>
> While it is true that our method incurs longer training time due to the use of RL, we would like to emphasize that this is a one-time cost. Once the expert vectors are trained, the inference cost of our method is only marginally higher (1st pass time is the extra cost), as we show in the table below (see also Table 3 in Sec 4.2). Moreover, the performance improvements achieved by our method, as demonstrated across almost all tasks, compensate for this upfront training cost.
>
> Additionally, a key advantage of our approach lies in the significantly fewer parameters and flexible composition capability. As we stated in the introduction, this not only reduces memory requirements but also makes our method inherently more suitable for constructing self-adaptive LLMs, where modularity and parameter efficiency are critical.
>
> | Task          | 1st Pass Time (s) | 2nd Pass Time (s) |
> |---------------|-------------------|-------------------|
> | MATH          | 42.64             | 321.19            |
> | HumanEval     | 2.76              | 14.28             |
> | ArcChallenge  | 13.40             | 28.51             |

---

> > ### Comment · Reviewer_MTB2 · 2024-11-25
> > **Official comment by Reviewer MTB2**
> >
> > The response resolves my concerns effectively, and I will keep my positive rating.

---

### Official Review · Reviewer_FCnJ · 2024-11-03

**Soundness:** 2
**Presentation:** 3
**Contribution:** 3
**Rating:** 6
**Confidence:** 4

**Summary:**

To address the challenge that traditional fine-tuning methods are often computationally intensive and static in their ability to handle diverse tasks, this paper proposes a self-adaptation framework that adapts LLMs by selectively adjusting the singular components of their weight matrices for unseen tasks at inference time. In the training stage for each task, it trains a vector to replace the singular values for each weight matrix. In the inference stage, it first extracts the task-related “singular vectors”, then makes inference with selected task-related “singular vectors”. Compared with Lora, experiments show the effectiveness of the proposed method.

**Strengths:**

1.	The proposed method of finetuning LLMs within eigenspaces is highly parameter efficient, as it only needs to train r = min(m, n) parameters for each weight matrix rather than $(m+n)\times r^\prime$ learnable parameters in Lora.
2.	The proposed method has the ability to mitigate catastrophic forgetting problem in continual learning of LLM, since it can adapt LLMs for unseen tasks at inference time by adjusting the singular components of their weight matrices.

**Weaknesses:**

1.	To demonstrate the effectiveness in parameter efficient finetuning domain, the experimental results are not sufficient as the paper only compares with Lora, which suffers from overfitting in their experiments. How about other parameter-efficient finetuning baselines, like Prefix Tuning, Prompt Tuning, DoRA, BOFT?
2.	The proposed method costs more time at inference time since it needs to make inference twice to get the final generated results. Particularly, for Few-shot adaptation, it needs to perform up to 100 CEM iterations, which costs significantly more time than normal inference.

**Questions:**

1.	For evaluation, is one model with three sets of “singular vectors” utilized for all three tasks(i.e., MATH, Humaneval, ARC-Challenge) for proposed methods? For Lora, three models are selected for three tasks, respectively?
2.	It looks that the proposed method has the ability to mitigate catastrophic forgetting problem in continual learning of LLM. The author may also consider exploring this direction to underline the contribution of this work further. (This is just a suggestion.)

---

> ### Author Response · Authors · 2024-11-24
> **Response to Reviewer FCnJ**
>
> We thank the reviewer for the constructive feedback. We are delighted to see that the reviewer agrees our method is highly parameter efficient and has the ability to mitigate catastrophic forgetting problems in continual learning of LLM. The following are our responses to the comments and questions, please see the details in the revised PDF.
>
> > To demonstrate the effectiveness in parameter efficient finetuning domain, the experimental results are not sufficient as the paper only compares with Lora, which suffers from overfitting in their experiments. How about other parameter-efficient finetuning baselines, like Prefix Tuning, Prompt Tuning, DoRA, BOFT?
>
> We have added comparisons with DoRA and IA$^3$  (suggested by Reviewer Kvfw), in addition, we have also conducted a hyperparameter sweep for LoRA and updated its results. These additional experimental results are reported in Tables 7 and 8. While we tried our best to follow the reviewer's instructions, due to the limited rebuttal time and computing resources we have, these comparisons are for Llama3-8B-Instruct. We will add complete comparisons in the camera-ready version of the paper.
>
> The following table gives a summary of the results (see also Tables 1, 2 and 7). Although most baselines are able to outperform the Llama3-8B-Instruct model after hyperparameter tuning, our method’s performance is the best in general and with the least trainable parameters.
>
>
> | Method                  | GSM8K    | MBPP-Pro | ARC-Easy | MATH     | Humaneval | ARC-Challenge |
> |-------------------------|----------|----------|----------|----------|-----------|---------------|
> | LLAMA3-8B-Instruct  | 75.89    | 64.65    | 88.59    | 24.54    | 60.98     | 80.63         |
> | + LoRA                  | 77.18    | **67.68**| 88.97    | 24.12    | 52.44     | 81.06         |
> | + IA$^3$                | 78.01    | **67.68**| 89.10    | 23.64    | 59.76     | 81.57         |
> | + DoRA                  | 78.09    | 64.65    | 89.14    | 24.44    | 52.44     | 81.14         |
> | + Ours                  | **79.15**| 66.67    | **89.56**| **25.47**| **62.99** | **82.61**     |
>
>
> > The proposed method costs more time at inference time since it needs to make inference twice to get the final generated results. Particularly, for Few-shot adaptation, it needs to perform up to 100
> iterations, which costs significantly more time than normal inference.
>
> We wish to point out that inference cost depends on the number of tokens generated. Although our method requires a 2-pass inference, the first pass merely generates a few tokens to decide the task identity and does not cause a large inference time increase (e.g., when using prompt-based or classifier-based adaptation, the first pass only generates one token). We provide a time measurement in the following to support this, we have also included this comparison in Sec 4.2.
>
> | Task          | 1st Pass Time (s) | 2nd Pass Time (s) |
> |---------------|-------------------|-------------------|
> | MATH          | 42.64             | 321.19            |
> | HumanEval     | 2.76              | 14.28             |
> | ArcChallenge  | 13.40             | 28.51             |
>
> Our CEM-based adaptation incurs the same 2nd pass time cost but eliminates the 1st pass overhead. Instead, it requires a one-time computational cost for evolutionary computations to determine the optimal combination of SVF-tuned vectors. This overhead depends on the settings (e.g., number of samples and CEM generations) and, in our experiments, is approximately 10 minutes for Arc-Challenge (CEM-light in Sec D), with similar times for MATH and HumanEval. While increasing the number of samples or CEM iterations can enhance performance, it also increases this one-time overhead, presenting a trade-off. To clarify this further, in addition to the ablation study on the number of examples presented in Table 6, we have added Section D to elaborate on this trade-off and its impact.
>
>
> > For evaluation, is one model with three sets of “singular vectors” utilized for all three tasks(i.e., MATH, Humaneval, ARC-Challenge) for proposed methods? For Lora, three models are selected for three tasks, respectively?
>
> Yes, the same base model (i.e., Llama3-8B-Instruct) with three sets of singular value scaling vectors is utilized for the three transferring-test tasks (i.e., MATH, Humaneval, ARC-Challenge). And yes, we trained separate LoRAs for these tasks, ensuring this is a fair comparison. This applies to all the new baselines we added in this revision too.
>
> > It looks that the proposed method has the ability to mitigate catastrophic forgetting problem in continual learning of LLM. The author may also consider exploring this direction to underline the contribution of this work further. (This is just a suggestion.)
>
> We thank the reviewer for the great suggestion, this is an important motivation for the work and we have added this in the introduction of the paper.

---

> ### Comment · Reviewer_FCnJ · 2024-11-29
>
> Thanks for the authors' efforts in providing the detailed response. Some of my concerns are addressed. The remaining one is about the experiments for baselines like Lora. Comparing Lora with the proposed method, we can see that Lora has more trainable parameters rather than scaling the singular vectors of the original parameter matrix, so I would expect Lora can fit better on the new target tasks like GSM8K, MBPP-Pro, ARC-Easy. Of course, with more trainable parameters, overly training may make Lora easier to suffer from overfitting. So, I'm wondering if the common early stopping strategy employed in the experiments for Lora baselines as did for SVF to mitigate overfitting problems?

---

> > ### Author Response · Authors · 2024-11-29
> > **Further Response to Reviewer FCnJ**
> >
> > We thank the reviewer for reading our responeses and we are glad to know that they have addressed some of the concerns. We provide answers to the follow-up questions below.
> >
> > > 1. One is about the experiments for baselines like Lora. Comparing Lora with the proposed method, we can see that Lora has more trainable parameters rather than scaling the singular vectors of the original parameter matrix, so I would expect Lora can fit better on the new target tasks like GSM8K, MBPP-Pro, ARC-Easy. Of course, with more trainable parameters, overly training may make Lora easier to suffer from overfitting. So, I'm wondering if the common early stopping strategy employed in the experiments for Lora baselines as did for SVF to mitigate overfitting problems?
> >
> > Yes, to ensure fair comparisons we employed a shared early stopping strategy. In particular, for all results, we evaluate the checkpoint at the iteration yielding the best performance on the validation sets of each task.
> >
> > We will be sure to state this more clearly right at the beginning of our Experiments section (line 317) in future revisions.
> >
> > > 2. When performing Singular Value Fine-tuning, how are the matrices $U, V^t$ calculated? If they are calculated by SVD, the rank r might be far less than $\min(m,n)$, leading to limited expressiveness.
> >
> > We used `torch.svd` (instead of `torch.svd_lowrank`) to get the decomposed matrices, and we confirmed the returned ranks were $\min(m,n)$. This is shown in our submitted source code, and also partially revealed in Figures 10 and 11 where we played with rank $r$ to show that low rank values $r\in\{16, 64, 256\}$ do not contain sufficient amount of information.
> >
> > However, we agree with the reviewer that even if SVD returned full ranks, because we only scale the singular values, SVF's expressiveness can be limited to the capability of the base model. We acknowledge this limitation and have discussed it in Section 5.
> >
> >
> > We hope the answers above address the reviewer's concerns. Please do not hesitate if you have more questions.

---

> > > ### Comment · Reviewer_FCnJ · 2024-11-29
> > >
> > > Thanks to the author for the prompt response. As most of my concerns have been addressed, I'm willing to raise my score to 6.

---

### Author Response · Authors · 2024-11-24
**General Response to Reviewers and AC**

We sincerely thank all the reviewers for their time and effort in evaluating our paper, as well as for their constructive and insightful feedback. We have carefully considered the comments and questions provided, and have made revisions to address these concerns. Additionally, we have conducted new experiments and updated the manuscript to reflect these changes (All changes are highlighted in blue text in the revised PDF).

Below, we provide a summary of the modifications made:

**Writing**
- Added text in the introduction to position LLM continual learning as a motivation (see Sec 1).
Added text in the method section to clarify the interaction and integration between components (see Sec 3.2).
- Added 2-pass inference time analysis (see Sec 4.2)
- Moved the ablation study from Appendix to the main text (see Sec 4.3)
- Added text in CEM-based adaptation to elaborate on efficiency (see Sec D) and scaling potentials (see Sec 5)
- Added the full name for SVF before it appears for the first time (see Sec 1)
- Modified the sentence about model merging in the conclusion to avoid confusion (see Sec 5)
- Modified Figure 4 and its caption to highlight the snapshots we used for evaluation (see Sec 4.1)
- Replaced the Reasoning curves in Figure 4 because we used a wrong log file to plot this in the original draft (the results reported are correct and unchanged)
- Modified the text suggested by reviewer Kvfw (see response to reviewer Kvfw)

**Experiments**
- Added comparison with IA3 (see Table 7), along with few-shot comparisons (see Table 8)
- Added comparison with DoRA (see Table 7)
- Conducted hyperparameter sweep for LoRA and updated its results (see Tables 1, 2 and 6)
- Added an extra setting of LoRA + next-token-prediction in ablation studies (see Table 4)

We hope that our responses and revisions satisfactorily address the reviewers’ concerns. If so, we kindly ask the reviewers to consider re-evaluating their ratings in light of these improvements.

---

### Meta-Review · Area_Chair_Posk · 2024-12-20

**Metareview:**

This paper introduces Transformer², a framework designed to enhance LLMs adaptability for unseen tasks in real-time. It employs a parameter-efficient method by dynamically adapting weight matrices using a two-pass mechanism to identify task properties and mix task-specific expert vectors trained with reinforcement learning. Transformer² demonstrates superior performance over traditional methods like LoRA in parameter efficiency and adaptability, applicable across various LLM architectures and modalities.

During the rebuttal period, the authors actively engaged in discussions with the reviewers, and the final scores were (6, 6, 6 ,6). Through discussions, the authors clarified and expanded on experimental baselines, computational costs, and the applicability of the method. Given the paper's novel approach, detailed analyses, and responsiveness to reviewer feedback, it offers significant value to the community and is recommended for acceptance.

The final decision is acceptance for the following reasons:

1. The idea of tuning singular values is novel and beneficial for reducing storage and computational costs.

2. The framework is adaptable to various LLM architectures and applicable to vision-language tasks, showcasing broad utility.

3. The paper is well-written and easy to follow. Extensive experiment results show the efficacy of the proposed methods.

**Additional Comments On Reviewer Discussion:**

This paper received consistent scores, all of which were 6 points, with positive ratings.

Reviewer FCnJ praises the method's parameter efficiency and its ability to mitigate catastrophic forgetting, with comparisons to IA3 and DoRA.

Reviewer MTB2 finds the computational cost of the reinforcement learning component justifiable.

Reviewer cbgq's suggestion to include ablation studies in the main text led to clearer insights.

Reviewer Kvfw's concerns about hyperparameters and reward function efficiency prompted a deeper explanation of SVF's role and integration of more expressiveness baselines.

AC agrees with the discussion results, and the authors have provided a list of their revisions and addressed the potential issues in the rebuttal. Therefore, the decision is to accept the paper.

---

### Decision · Program_Chairs · 2025-01-22

Accept (Poster)